



# A Bayesian Ensemble Data Assimilation to Constrain Model Parameters and Land Use Carbon Emissions

Sebastian Lienert[1,2] and Fortunat Joos[1,2]

[1]Climate and Environmental Physics, Physics Institute, University of Bern, Bern, Switzerland
[2]Oeschger Centre for Climate Change Research, University of Bern, Bern, Switzerland.

*Correspondence to:* Sebastian Lienert (lienert@climate.unibe.ch)

**Abstract.**

A dynamic global vegetation model (DGVM) is applied in a probabilistic framework and benchmarking system to constrain uncertain model parameters by observations and to quantify carbon emissions from land-use and land-cover change (LULCC). Processes featured in DGVMs include parameters which are prone to substantial uncertainty. To cope with these uncertainties Latin Hypercube Sampling (LHS) is used to create a 1000-member perturbed parameter ensemble, which is then evaluated with a diverse set of global and spatio-temporaly resolved observational constraints. We discuss the performance of the constrained ensemble and use it to formulate a new best-guess version of the model (LPX-Bern v1.4). The observationally constrained ensemble is used to investigate historical emissions due to LULCC ($E_{\mathrm{LUC}}$) and their sensitivity to model parametrization. We find a global $E_{\mathrm{LUC}}$ estimate of 158 (108, 211) PgC (median and 90% confidence interval) between 1800 and 2016. We compare $E_{\mathrm{LUC}}$ to other estimates both globally and regionally. Spatial patterns are investigated and estimates of $E_{\mathrm{LUC}}$ of the ten countries with the largest contribution to the flux over the historical period are reported. We consider model versions with and without additional land-use processes (shifting cultivation and wood harvest) and find that the difference in global $E_{\mathrm{LUC}}$ is on the same order of magnitude as parameter induced uncertainty and in some cases could potentially even be offset with appropriate parameter choice.

## 1 Introduction

Due to constraining atmospheric $CO_2$ concentrations and the relatively well known $CO_2$ sink in the ocean it follows that about a fifth of anthropogenic $CO_2$ emissions is assimilated by the terrestrial biosphere (Ciais et al., 2013). However, the partitioning of this land-atmosphere flux to effects from human-induced land-use and land-cover change (LULCC) and the transient change of the residual terrestrial sink remains highly debated (Schimel et al., 2015). It is estimated that approximately a third of the cumulative anthropogenic $CO_2$ emissions in the industrial period stem from the effects of LULCC (Arneth et al., 2017; Brovkin et al., 2013; Gerber et al., 2013; Houghton and Nassikas, 2017; McGuire et al., 2001; Mahowald et al., 2017; Pongratz and Caldeira, 2012; Sitch et al., 2015; Strassmann et al., 2008; Stocker et al., 2017, 2014; Peng et al., 2017). A better understanding of the mechanisms of the historical terrestrial carbon cycle is vital for more accurate future projections of the global carbon

cycle and climate. In addition,a better understanding of the residual terrestrial sink can also help to improve our understanding of the terrestrial carbon cycle of the past, unperturbed by human influence.

Dynamic Global Vegetation Models (DGVMs) can be used to assess the contribution of LULCC to the terrestrial carbon budget (Le Quéré et al., 2016). In addition to uncertain prescribed LULCC forcings and processes in DGVMs, the parametriza-

tion of those processes is subject to further uncertainties. We use a Monte-Carlo-like data assimilation approach (Steinacher et al., 2013; Steinacher and Joos, 2016; Battaglia and Joos, 2017) to sample 15 key model parameters and construct a 1000-member model ensemble to investigate this parameter related uncertainty in the DGVM LPX-Bern. Furthermore, we establish a new reference version of the model.

The assimilation of observations is an integral part of model development. Various approaches to incorporate constraining

data exist, such as variational approaches minimizing a cost function using the adjoint of the model (Houweling et al., 1999; Kato et al., 2013; Kaminski et al., 2013) or the use of ensemble Kalman filters (Houweling et al., 1999; Lorenc, 2003; Gerber and Joos, 2013; Stöckli et al., 2011; Ma et al., 2017). However since these methods operate sequentially, the metric and relative weighting of observational products have to be chosen beforehand, which can prove difficult if multiple observational data sets are to be assimilated simultaneously. The assessment of the performance of a given model version using observational

benchmarks has also been actively discussed in the literature (Hoffman et al., 2017; Peng et al., 2014; Kelley et al., 2013; Luo et al., 2012; Blyth et al., 2011; Randerson et al., 2009) and different frameworks have been proposed. Here we employ the Latin Hypercube Sampling (LHS) (McKay et al., 1979) approach, as used successfully in previous studies (Steinacher et al., 2013; Battaglia et al., 2016; Steinacher and Joos, 2016; Battaglia and Joos, 2017; Zaehle et al., 2005). It allows simultaneous stratified sampling of a range of parameters, given an appropriate prior parameter distribution, while offering the opportunity to change

evaluation metrics a posteriori, thus enabling a sensible incorporation of multiple observational constraints. By improving the prior distribution iteratively it is possible to reasonably capture observations while considering trade-offs between the different targets. Additionally, this approach not only yields a best-guess of parameter values but also contains information about the associated uncertainties. A drawback of this technique is that it is not possible to increase the size of the ensemble after the initial sampling and if the range of the prior distribution is too large the algorithm has decreased computational efficiency.

The selection of observational targets is vital to a successful assimilation of observational data. In order to constrain the contemporary carbon cycle, 14 data products are used, ranging from global inventories of carbon (Ciais et al., 2013) to spatially resolved satellite estimates of photosynthetically absorbed radiation (Gobron et al., 2006). The goal of the data set selection process was to have observations capturing the magnitudes of fluxes and inventories in the carbon cycle, as well as its transient response to anthropogenic perturbance. The inclusion of further independent observations in this framework would further

serve to reduce parameter uncertainty.

While the land-atmosphere carbon flux can be constrained by the other components of the global carbon cycle, the contribution of LULCC, and in turn the implied terrestrial carbon sink, are highly uncertain. Efforts to fill this knowledge gap have been made using bookkeeping approaches (Houghton et al., 2012; Hansis et al., 2015; Houghton and Nassikas, 2017) and bottom-up modeling approaches using DGVMs (McGuire et al., 2001; Stocker et al., 2014; Wilkenskjeld et al., 2014; Sitch

et al., 2015). Bookkeeping models can offer valuable information on the magnitude of regional and global LULCC emissions




($E_{\mathrm{LUC}}$), but they typically rely on time-invariant estimates of carbon densities and thus neglect the direct impact of climate change on vegetation. Observational data on carbon densities and response of the vegetation to LULCC effects can be directly incorporated in bookkeeping models. In contrast, DGVM model studies are able to produce highly resolved spatial results and consider changes to vegetation structure due to anthropogenic perturbance, but DGVMs have large uncertainties due to

differences in process modeling and parametrization. Additionally, a number of LULCC processes are often neglected, such as the effect of gross land-cover transitions (shifting cultivation), management (wood harvest) or erosion. Studies investigating these processes generally have found that the inclusion of those processes leads to an increase in $E_{\mathrm{LUC}}$ (Arneth et al., 2017; Wilkenskjeld et al., 2014; Stocker et al., 2014). On the other hand, neglected processes such as human-induced erosion can have the opposite effect and reduce net $E_{\mathrm{LUC}}$ (Kosmas et al., 2007; Billings et al., 2010; Hoffmann et al., 2013; Wang et al.,

2017). The effect of parameter uncertainty on these estimates is often only considered indirectly in the intercomparison of models. Here we investigate a parameter ensemble of a single DGVM, constrained by observation and provide direct estimates of parameter induced uncertainties in LULCC estimates. These uncertainties are put into context by investigating the effect of additional LULCC processes, such as shifting cultivation and wood harvest, as already investigated in previous studies (Stocker et al., 2014; Wilkenskjeld et al., 2014; Shevliakova et al., 2009). We rely here on the LUH2 v2h (Hurtt et al.) land-cover data

to force the DGVM LPX-Bern v1.4.

## 2   Methods

### 2.1   LPX-Bern

The Land Surface Processes and eXchanges (LPX-Bern) model (Spahni et al., 2013; Stocker et al., 2013; Keller et al., 2017) is a Dynamic Global Vegetation Model (DGVM) based on the Lund-Potsdam-Jena (LPJ) model (Sitch et al., 2003). It features

coupled nitrogen, water and carbon cycles and distinguishes between different types of prescribed land-use classes: natural vegetation, peatland, cropland, pasture and urban land. The vegetation composition for a given land-use class is determined dynamically. Different plant functional types (PFTs), with given bioclimatic limits, compete for resources. Here 9 tree PFTs and 2 herbaceous PFTs are used to describe natural vegetation, the same two generic herbaceous PFTs grow on pasture and cropland, and two moss PFTs, two flood tolerant tropical PFTs, and a flood-tolerant herbaceous PFT grow on peatlands.

Two different configurations are used to treat the transition between different classes of land-use. The simpler implementation adjusts the fractional land-use cover at the end of each year such that the prescribed area fractions are achieved, this computationally efficient configuration is referred to as net land-use. The more advanced gross land-use implementation also includes effects of shifting cultivation and wood management by prescribing all the transitions between different land-use classes and harvested wood (Stocker et al., 2014; Strassmann et al., 2008). Furthermore, it includes an additional land-use

class, the so-called secondary forest, natural vegetation growing on abandoned pasture or cropland. A major drawback of this scheme is the significantly increased computational cost. Additionally, the implementation of gross land-use in LPX-Bern in the current version does not allow for the simultaneous simulation of peatlands. For both schemes a fraction $oc_{frac}$ of the crops above-ground biomass is directly oxidized to the atmosphere, simulating crop harvest. 75% of heartwood and sapwood





biomass production from forest conversion is assigned to decaying product pools, the remaining 25% are respired directly to the atmosphere as assumed harvest losses. Associated root and leaf mass are transferred to an below and above ground litter pool respectively. The biomass in the product pools is evenly split in a long-lived (mean lifetime 20 years) and a short-lived (mean lifetime 2 years) pool. In the gross LULCC setup biomass is harvested according to the prescribed forcing and the resulting heartwood is assigned to product pools using the same allocation rules as before.

## 2.2 Model setup and spinup

The model is run on a 1° x 1° global grid and forced with CRU TS3.23 climate data (Harris et al., 2014) and global atmospheric $CO_2$ concentration from ice core reconstructions (Meure et al., 2006; Joos and Spahni, 2007) and direct atmospheric measurements after 1958 (Tans and Keeling). The Land-Use Harmonization LUH2 v2h (Hurtt et al.) estimates for land-use patterns and transitions are prescribed to the model. Additionally nitrogen deposition (Lamarque et al., 2013) and fertilization (Zaehle et al., 2011) and the extent of northern hemisphere peatlands (Tarnocai et al., 2009) are prescribed. As described in section 2.3 we use an ensemble approach featuring 1000 simulations with different parameters. All ensemble members share a 1500 year spin-up run to pre-industrial conditions, using the median parameter values. To ensure the equilibration of each member an additional 300 year individual spin-up run, featuring an analytical equilibration of the soil carbon pools after 100 years, is performed. The model is then run transiently from 1800 to 2014 with recycled climate data (years 1901-1930) in the 19th century.

## 2.3 Sampling

The model parameter space is sampled using Latin Hypercube Sampling (LHS) (McKay et al., 1979) to create an ensemble of model configurations and assess model uncertainty. LHS is a stratified sampling method using chosen prior parameter distribution to generate a parameter ensemble of a given size. In contrast to most Monte Carlo sampling techniques, the sampling is independent of the model metrics, allowing to modify the metrics without large computational effort. A drawback of this sampling strategy is that it is not possible to increase the size of the ensemble after the initial sampling.

The selected sampling parameters and their assumed prior distribution are given in Table 1. The parameters were selected for their importance in various aspects of the model, 10 of the 15 parameters were also used by (Steinacher et al., 2013). The parameter selection was further guided by an earlier study by Zaehle et al. (2005), investigating the relative importance of 36 parameters in the LPJ model. The fraction of photosynthetically active radiation, $\alpha_a$, the intrinsic quantum efficiency of $CO_2$ uptake in C3 plants, $\alpha_{C3}$ and $\theta$ the rubisco co-limitation shape parameter are of primary importance for the photosynthetic carbon assimilation. $g_m$ and $\alpha_m$ are parameters in the empiric water demand calculation and have a direct impact on the hydrological cycle and consequently also the carbon assimilation. The sapwood-heartwood turnover time, $\tau_{sapwood}$, the maximum mortality parameter, $mort_{max}$, and the ratio between leaf area and sapwood area, $k_{la:sa}$, are vital for the allocation of the carbon to the different vegetation pools and thus also the overall vegetation carbon pool size. The fraction of the flux leaving the litter pools that is respired to the atmosphere directly and entering the slow soil pool, $f_{atm}$ and $f_{slow}$ influence soil and litter carbon inventories. These pools are further controlled by the temperature sensitivity of heterotrophic respiration $E_{0,hr}$,





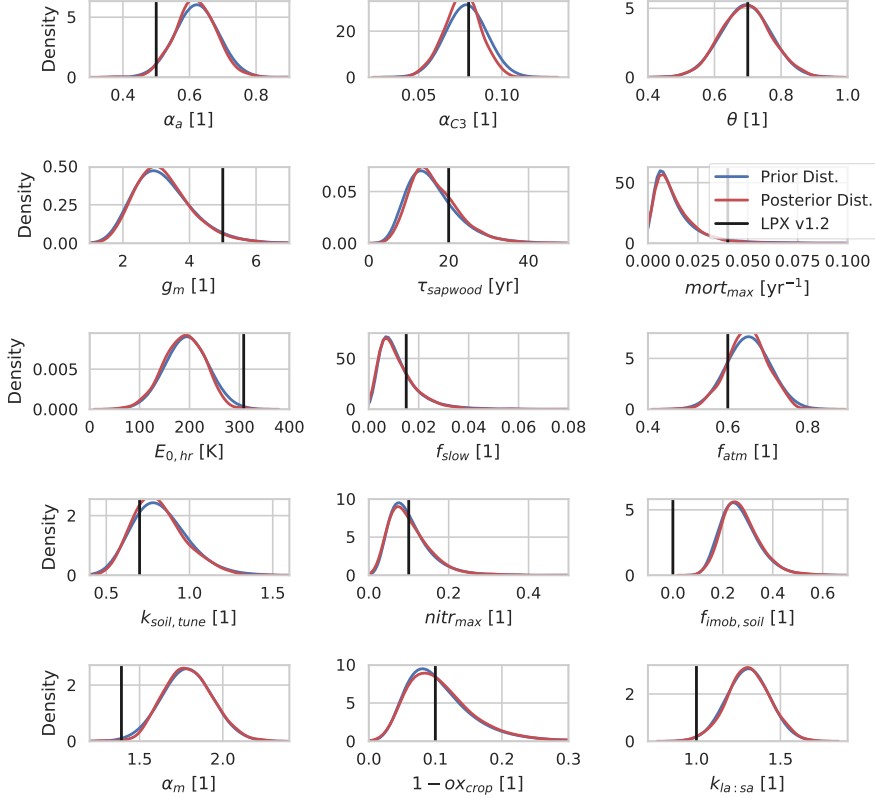

**Figure 1.** Kernel density estimations of the prior probability distribution (blue) and the posterior probability distribution of the constrained ensemble ($M_{net,net}$). The prior distribution was improved iteratively, resulting in near convergence of prior and posterior distribution. Vertical black bars indicate the parameter values used in LPX-Bern v1.2.

which is of special significance under changing climate, and a scaling factor for soil decomposition $k_{soil,tune}$, affecting the residence time of both the fast and the slow soil carbon pool. By using factorial simulations two important parameters for the nitrogen cycle were identified, the maximum nitrification rate, $nitr_{max}$, and the fraction governing immobilization of mineral nitrogen in the soil, $f_{imob,soil}$. Finally, the oxidation rate of crops $ox_{crop}$, representing the harvest of biomass on croplands, is

5  directly linked to emissions from human land-use.

Each prior parameter distribution is sampled from either a normal or log-normal distribution. The prior distributions were chosen starting from literature-based plausible ranges. Then these initial priors were refined iteratively by applying the Bayesian approach, as described in this and the next section, in sequence using multiple ensembles and additional factorial simulations for selected parameters. The prior distributions (Table 1) of the featured ensemble are based on the 95% confidence interval

10  of the constrained posterior parameter distribution of an earlier ensemble run. While during ensemble revisions some metrics





**Table 1.** Description of sampling parameters with values for LPX v1.2 and the new best guess version v1.4. If not otherwise indicated parameters are unitless.

| Paramter | Description | LPX v1.2 | LPX v1.4 |
|---|---|---|---|
| $\alpha_a$ | Fraction of PAR assimilitated at ecosystem level relative to leaf level | 0.5 | 0.6175 |
| $\alpha_{C3}$ | Intrinsic quantum efficiency of CO_2 uptake in C3 plants | 0.07 | 0.07660 |
| $\theta$ | Co-limitation shape parameter | 0.7 | 0.6937 |
| $g_m$ | Canopy conductance scaling parameter for water demand calculation | 3.24 | 3.120 |
| $\alpha_m$ | Priestley-Taylor coefficient in water demand calculation | 1.394 | 1.786 |
| $\tau_{sapwood}$ | Sapwood-heartwood turnover time [yr] | 20 | 15.33 |
| $k_{la:sa}$ | Allometric scaling parameter: leaf area to sapwood area | 1.0 | 1.310 |
| $mort_{max}$ | Asymptotic maximum in mortality equation [$\text{yr}^{-1}$] | 0.01 | 0.01016 |
| $E_{0,hr}$ | Temperature sensitivity of heterotrophic respiration [K] | 308.56 | 190.16 |
| $f_{atm}$ | Fraction of litter entering atmosphere directly | 0.6 | 0.6503 |
| $f_{slow}$ | Fraction of litter entering slow soil pool | 0.015 | 0.009512 |
| $k_{soil,tune}$ | Tuning factor for soil decay | 0.7 | 0.7965 |
| $nitr_{max}$ | Maximum nitrification rate | 0.1 | 0.09096 |
| $f_{imob,soil}$ | Nitrogen imobilization in soil | 0.0 | 0.2639 |
| $1 - ox_{crop}$ | Fraction of direct oxidation of leaf turnover on cropland | 0.1 | 0.09920 |

have changed and forcing files were updated, the result of this iterative procedure leads to a near convergence of the prior and the posterior parameter distribution for the final ensemble (Fig. 1).

## 2.4 Skill scores

The performance of the model ensemble is evaluated using a set of observational constraints as listed in Table 2. The model-
5 data discrepancy for a given observational data set $i$ and model run is estimated by the relative Mean Squared Error ($\text{MSE}^i_{rel}$)

$$\text{MSE}^i_{rel} = \sum_j w_j \frac{(X_j^{mod,i} - X_j^{obs,i})^2}{\sigma^2}. \tag{1}$$

$w_j$ are the normalized weights of the data points j, which in the case of gridded data sets correspond to the grid cell area. $X_j^{mod,i}$ and $X_j^{sim,i}$ correspond to the modeled and observed data points for constraint $i$ respectively. In accordance with (Schmittner
10 et al., 2009) and (Steinacher et al., 2013) the combined error $\sigma^2$ is approximated by the model-data variance of the model member with the smallest $\text{MSE}^i_{rel}$ of the ensemble. As a consequence, the smallest possible $\text{MSE}^i_{rel}$ using this approximation is one. If the observational error is known and larger than the variance, it is instead used as an estimate for the combined error, allowing a theoretical minimum $\text{MSE}^i_{rel}$ of zero.





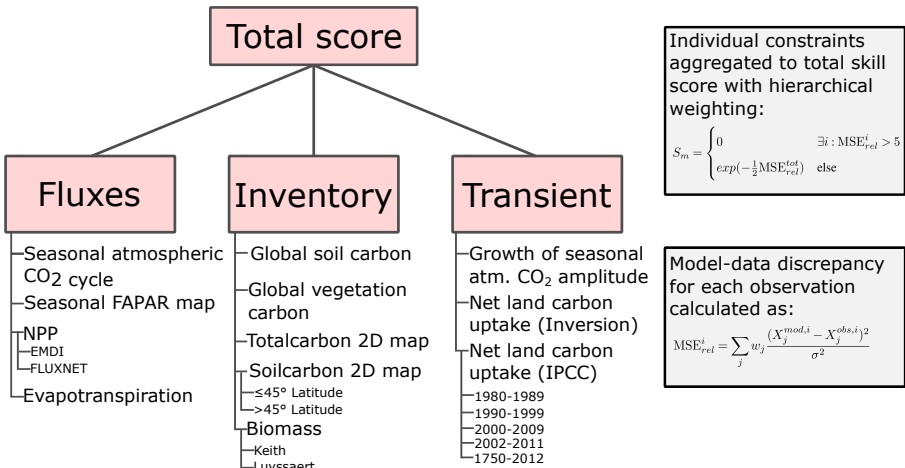

**Figure 2.** Hierarchical weighting scheme to aggregate the relative mean squared error of individual observational constraints to a total error which is then mapped to a total skill score.

The $\mathrm{MSE}_{rel}$ of all individual observational constraints is aggregated to a total error $\mathrm{MSE}_{rel}^{tot}$ with a hierarchical weighting scheme shown in Fig. 2 and translated to a skill score $S_m$ for each ensemble member $m$. We require that $\mathrm{MSE}_{rel}$ is smaller than five for each of the individual observational data sets; otherwise $S_m$ is set to 0.

$$S_m = \begin{cases} 0 & \exists i : \mathrm{MSE}_{rel}^i > 5 \\ exp(-\frac{1}{2}\mathrm{MSE}_{rel}^{tot}) & \text{else} \end{cases} \tag{2}$$

The size of the ensemble is further reduced by excluding runs with low skill scores, such that the remaining 667 runs have 99 % of the cumulative skill score $\sum_m S_m$ of all runs. The maximum achievable skill score is not 1 since it would correspond to a $\mathrm{MSE}_{rel}^{tot}$ of 0, which is not achievable due to the approximation for the combined error, used in some of the observational constraints. We did not renormalize skill score to a scale between 0 and 1.

The skill score weighted normalized histogram of a quantity of interest can be interpreted as a probability density function.
The skill weighted median and confidence interval of a given quantity is then determined by transforming the histogram to a discrete cumulative density function using a cumulative sum and approximating the desired quantiles by a first-order interpolation.

### 2.5    Observational constraints

The calculation of the $\mathrm{MSE}_{rel}$ requires the model and observational data to conform to the same structure. In the following, the
required pre-processing will be outlined briefly. The seasonality of the fraction of absorbed photosynthetically active radiation (FAPAR) as simulated in the model is compared to a satellite-derived product (Gobron et al., 2006) which was regridded to the model grid and the MSE is calculated from the averaged monthly fields in the measurement period.



**Table 2.** Observations used to constrain the model ensemble

| Category | Variable | Description | Reference |
|---|---|---|---|
| Fluxes | Seasonal atm. $CO_2$ | Seasonal cycle at nine sites | (GLOBALVIEW-CO2, 2013) |
| Fluxes | NPP | Estimates of the 81 Multi-Biome Class A field measurements | (Olson et al., 2013) |
| Fluxes | NPP | Estimates of NPP at ∼140 FLUXNET sites | (Luyssaert et al., 2009, 2007) |
| Fluxes | fAPAR | SeaWiFS satelite product, regridded to model resolution | (Gobron et al., 2006) |
| Fluxes | Evapotranspiration | Merged evapotranspiration synthesis product from the LandFlux-EVAL | (Mueller et al., 2013) |
| Inventory | Total Carbon | Global distribution of total ecosystem carbon | (Carvalhais et al., 2014) |
| Inventory | Soil Carbon | Global distribution of total soil carbon | (Carvalhais et al., 2014) |
| Inventory | Vegetation Carbon | Biomass estimates at ∼140 FLUXNET sites | (Luyssaert et al., 2009, 2007) |
| Inventory | Vegetation Carbon | Biomass estimates at 136 sites | (Keith et al., 2009) |
| Inventory | Global Soil Carbon | Global inventory $1950 \pm 450$ GtC | (Ciais et al., 2013) |
| Inventory | Global Vegetation Carbon | Global inventory $550 \pm 100$ GtC | (Ciais et al., 2013) |
| Transient | Growth of $CO_2$ amplitude | Growth of seasonal atmospheric $CO_2$ amplitude at four sites | (GLOBALVIEW-CO2, 2013) |
| Transient | Land uptake (Inversion) | Global land uptake from atmospheric inversion | (Boden et al., 2017; Battaglia and Joos, 2017) |
| Transient | Land uptake (IPCC) | Global land uptake in five periods | (Ciais et al., 2013) |

The modeled total and soil carbon distribution between 1982 and 2005 are compared to a data set based on observations (Carvalhais et al., 2014), regridded to the model resolution. The soil carbon map is divided into low and high latitudes regions in order to avoid potential biases from peat areas with very high soil carbon content.

For site level observed NPP (Multi-Biome NPP (Olson et al., 2013) and FLUXNET v3.1 (Luyssaert et al., 2009, 2007)),

5    the site measurements are compared to the averaged modeled NPP of natural vegetation between 1931 and 1997 of the corresponding model grid cell. If multiple measurements are contained in the same grid cell they are averaged. Similarly, the site level measurements of biomass carbon (Keith et al., 2009; Luyssaert et al., 2009, 2007) are compared to the modeled natural vegetation carbon, averaged between the periods 1950-2000 and 1931-1997 respectively. The biomass carbon of Luyssaert et al. (2009) is obtained by using a carbon to organic matter conversion factor of 0.475.

10    The TM2 (Kaminski et al., 1999), a global atmospheric tracer model was used to translate the gridded land-atmosphere flux to local anomalies in atmospheric $CO_2$. 9 sites from the GLOBALVIEW-CO2 database (GLOBALVIEW-CO2, 2013) were selected and the annual offset corrected seasonality of $CO_2$ in the period of 1980-2013 was compared. The influence of sea-air carbon exchange on the seasonal cycle and trend in atmospheric $CO_2$ are taken into account. This is done by prescribing net



sea-to-air fluxes as simulated by the Bern3D model (standard setup) (Battaglia and Joos, 2017; Roth et al., 2014; Ritz et al., 2011). The growth of the seasonal amplitude at a subset of four sites with high seasonality was used as a further constraint.

The modeled mean annual evapotranspiration between 1989-2005 was compared to the LandFLUX-EVAL evapotranspiration data product (Mueller et al., 2013).

The global terrestrial carbon flux is constrained by an inversion, for which the global atmospheric $CO_2$ concentration, the median of an ensemble of simulated ocean-atmosphere fluxes (Battaglia and Joos, 2017), consistent with other estimates (Khatiwala et al., 2013; DeVries, 2014), and an inventory of anthropogenic $CO_2$ emissions (Boden et al., 2017) were used. The combined error in Equation 1 is estimated by propagating the 90% confidence interval of ocean-atmosphere fluxes and assuming a 5% uncertainty for the anthropogenic emissions (Ballantyne et al., 2015).

The estimates of global soil and vegetation carbon as given by IPCC (Ciais et al., 2013) are used as a global constraint. The observation-based estimates are compared to the average soil and vegetation carbon over the whole industrial period. Additionally, the estimates for the global land-atmosphere flux in the periods 1970-1979, 1980-1989, 1990-1999, 2000-2009 and 2002-2011, are compared to the simulated land-atmosphere fluxes over the same period. Since the model simulation starts only in the year 1800, the estimated land-atmosphere flux over the industrial period from 1750-2011 is compared with the

model by approximating the flux of the period 1750-1800 with 1801-1850. For all global constraints, the uncertainties reported by IPCC are used as an estimate for the combined error in Equation 1.

### 2.6   Definition of Land-Use emissions and the setup of three model ensembles

To quantify emissions from LULCC a second simulation featuring a time-invariant pre-industrial land-cover distribution and nitrogen fertilization is run for every ensemble member. In accordance with the TRENDY model intercomparison ((Sitch

et al., 2015)), we define the emissions from LULCC as the difference of the change in carbon in the reference and fixed LULCC simulation. The change of carbon in the land system is calculated from the cumulative net biome production (NBP) including emissions from product pools. Since the additional simulations with fixed LULCC feature transient $CO_2$ and climate forcing, the direct impact of climate change and increasing $CO_2$ on $E_{\mathrm{LUC}}$ are considered, however unlike in coupled models (Strassmann et al., 2008) physical and biogeochemical feedbacks of LULCC on the climate are neglected. We refer to the

literature (Strassmann et al., 2008; Pongratz et al., 2014; Stocker and Joos, 2015) for further discussion of differences in the definition of land-use fluxes.

$E_{\mathrm{LUC}}$ is investigated using three different ensemble configurations. $M_{net,net}$ labels the standard model version featuring only net LULUC transitions. $M_{gross,net}$ and $M_{gross,gross}$ feature modules for shifting cultivation and wood harvest (gross land-use) but lack northern peatlands due to technical limitations. $M_{gross,net}$ reuses the skill scores calculated for $M_{net,net}$ and

$M_{gross,gross}$ features skill scores calculated on the basis of the gross land-use configuration.

For the $M_{net,net}$ ensemble and the $M_{gross,net}$ ensemble, the priors of the model parameters were improved iteratively during the development of our benchmark system. Consequently, the solutions for the model parameters and associated model outcomes converge. For example, the prior and the posterior probability distribution of the sampled parameters are nearly identical (Fig. 1). This provides strong support that an optimal solution for the sampled parameters has been found for the applied model





structure and observational constraints. In contrast, the parameters of the M$_{gross,gross}$ ensemble were not improved iteratively, given the computational cost, and prior and posterior solutions do not converge.

## 3 Results

### 3.1 Land-Use Emissions

The magnitude of emissions of carbon due to changes in LULCC is hard to quantify and subject to large uncertainties (Li et al., 2017; Arneth et al., 2017; Houghton and Nassikas, 2017; Pongratz et al., 2011; Roman-Cuesta et al., 2016; Stocker et al., 2014; Strassmann et al., 2008; Wang et al., 2017). The use of an ensemble framework allows us to quantify both the magnitude and the uncertainty of land-use emissions in a model due to parameter spread. Following the procedure outlined in the method section, $E_{\mathrm{LUC}}$ is computed for every ensemble member. In this section, we first present $E_{\mathrm{LUC}}$, total land-atmosphere fluxes and

the residual land carbon sink on a global scale for the three ensemble configurations and then further analyze spatial patterns and regionally aggregated estimates.

### 3.1.1 Global Fluxes

Global aggregates of skill weighted median NBP, $E_{\mathrm{LUC}}$, residual terrestrial sink and their respective cumulative fluxes, including a 90 % confidence interval as an estimate for model parameter uncertainty, are shown in Fig. 3. For the standard model

configuration M$_{net,net}$, featuring net land-use, the total change in land carbon (i.e. cumulative NBP) is a release of 24.2 PgC from 1860 to 1960 and an uptake of 25.5 PgC from 1961 to 2016. The standard deviation of NBP increases from 0.8 PgC/yr between 1860 and 1960 to 1.2 PgC/yr in the latter period. The change in total carbon is discussed in more detail in section 3.3. $E_{\mathrm{LUC}}$ is positive throughout the whole industrial period, i.e. a source of carbon to the atmosphere. A temporary maximum of emissions is reached in the 1950s followed by relatively constant emissions until the 2000s where the emissions increase with

enhanced variability. The cumulative emissions from 1860 to 2016 amount to 98.0 PgC. The residual terrestrial sink, computed as the difference between NBP and $E_{\mathrm{LUC}}$, shows a similar pattern of variability as NBP. While the residual terrestrial sink flux is negative in some years, the cumulative residual terrestrial sink generally increases steadily and amounts to 98.9 PgC between 1860 and 2016.

  In addition to the standard model configuration a second ensemble of a model configuration M$_{gross,net}$ featuring modules

for shifting cultivation and wood harvest (gross land-use) is employed. By using the skill scores M$_{net,net}$, the parametrization remains identical allowing to compare the pure mechanistic difference between the two versions. The difference in $E_{\mathrm{LUC}}$ between the net and gross land-use configuration is most pronounced in the second half of the 20th century and amounts to 43.7 PgC between 1860 and 2016. The gross land-use ensemble simulates on average 0.41 GtC yr$^{-1}$ more emissions due to LULUC between 1950 and 2016. This result is compatible with the earlier study by (Stocker et al., 2014), which investigated

land-use-change using an earlier version of LPX-Bern with a single parameter configuration. The residual terrestrial sink shows





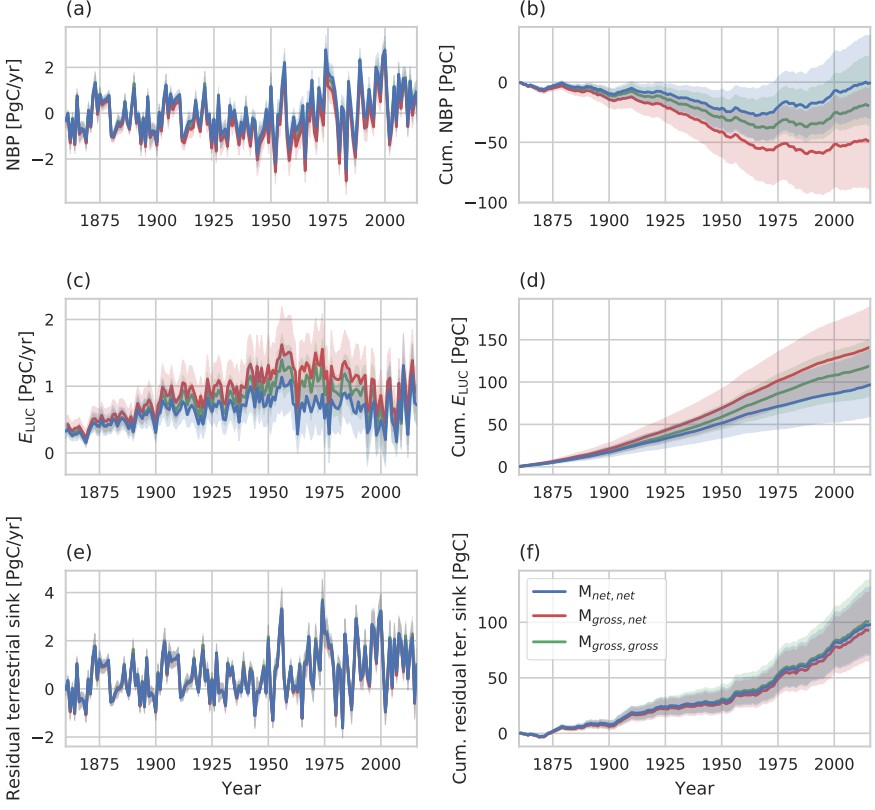

**Figure 3.** Skill weighted median Net Biome Production (NBP) (a), emissions due to LULUC $E_{\mathrm{LUC}}$ (c) and the residual terrestrial sink flux (e) and their respective cumulative fluxes (b,d,f) for the constrained ensemble with net land-use $M_{net,net}$ (blue), additional gross land-use processes $M_{gross,net}$ (red) and gross land-use with skill scores recalculated $M_{gross,gross}$ (green). The shading corresponds to the 90% confidence interval.

as expected a near identical behavior in the two versions. The resulting total change in land carbon is negative, with a slight uptake of carbon after 1980.

A third model configuration $M_{gross,gross}$ is obtained by recalculating the skill scores from the gross land-use results. As described in section 2.6, the priors of the $M_{gross,gross}$ were not improved iteratively to yield convergence between prior and pos-

5   terior solutions. This leaves only 200 runs in $M_{gross,gross}$ in contrast to the 667 runs in $M_{net,net}$ and consequently $M_{gross,net}$. In addition, several important benchmarks such as vegetation carbon density are not simulated as well in $M_{gross,gross}$ compared to $M_{net,net}$ and $M_{gross,net}$. Since NBP is constrained by observations, median cumulative NBP from 1860 to 2016 is only 17.5 PgC smaller in the $M_{gross,gross}$ than in the $M_{net,net}$ ensemble. Surprisingly $E_{\mathrm{LUC}}$ is only 21.4 GtC higher over the period from 1860 to 2016 for $M_{gross,gross}$ than for the standard version $M_{net,net}$. Why are $E_{\mathrm{LUC}}$ emissions so similar between

10   these two ensemble versions with net and gross transitions? The residual sink is relatively insensitive to parameterization in





**Table 3.** Comparison of the skill weighted median emissions due to Land-use-change of the two constrained LPX parameter ensembles (90% confidence intervals in brakets) to the bookkeeping method and DGVM model ensemble of (Le Quéré et al., 2016). The uncertainty in the DGVM multi-model ensemble is given by the standard deviation across model members, for the bookkeeping method a best value judgement on the uncertainty of $\pm 0.5$ PgC yr$^{-1}$ is provided.

| | Mean $E_{\text{LUC}}$ [PgC yr$^{-1}$] | | | | |
| | 1960-1969 | 1970-1979 | 1980-1989 | 1990-1999 | 2000-2009 |
| --- | --- | --- | --- | --- | --- |
| LPX-Bern $M_{net,net}$ | 0.70 (0.33, 1.04) | 0.69 (0.30, 1.06) | 0.75 (0.40, 1.07) | 0.55 (0.22, 0.83) | 0.52 (0.21, 0.78) |
| LPX-Bern $M_{gross,net}$ | 1.22 (0.78, 1.64) | 1.25 (0.77, 1.71) | 1.19 (0.77, 1.57) | 0.93 (0.54, 1.28) | 0.74 (0.41, 1.05) |
| LPX-Bern $M_{gross,gross}$ | 1.02 (0.65, 1.32) | 1.04 (0.65, 1.37) | 0.99 (0.66, 1.27) | 0.74 (0.37, 1.05) | 0.59 (0.26, 0.87) |
| GCP2016 Bookkeeping | $1.5 \pm 0.5$ | $1.3 \pm 0.5$ | $1.4 \pm 0.5$ | $1.6 \pm 0.5$ | $1.0 \pm 0.5$ |
| GCP2016 DGVMs | $1.2 \pm 0.3$ | $1.2 \pm 0.3$ | $1.2 \pm 0.2$ | $1.1 \pm 0.2$ | $1.3 \pm 0.3$ |

LPX and the version with gross skill scores only has a moderately larger residual sink uptake of 7.8 PgC in the considered period, largely caused by a downward adjustment of the parameter $E_{0,hr}$ governing the temperature dependency in heterotrophic respiration to a median value of 151 K (190 K in $M_{net,net}$; Table 1). In $M_{gross,gross}$, global vegetation carbon inventory is only 417 PgC (average over the industrial period) compared to 468 GtC in the $M_{net,net}$ ensemble. The observational con-
straints for the net land carbon sink (Fig. 2, Table 2) are apparently better approximated for a smaller vegetation carbon stock in $M_{gross,gross}$. Vegetation carbon inventory is underestimated by about 20% compared to the observational benchmarks. The smaller vegetation carbon stock in $M_{gross,gross}$ leads to smaller $E_{\text{LUC}}$ anything else kept equal. In addition, the amount of carbon harvested ($ox_{crop}$) is reduced from 90% in the standard $M_{net,net}$ ensemble to 83% in the $M_{gross,gross}$ ensemble. As a result of these two adjustments, $E_{\text{LUC}}$ is smaller in the $M_{gross,gross}$ than in the $M_{gross,net}$ ensemble. If the relative impor-
tance of the land-atmosphere observational constraints is increased, the difference in $E_{\text{LUC}}$ of $M_{gross,gross}$ and $M_{net,net}$ is decreased even further.

$E_{\text{LUC}}$ as simulated by LPX-Bern is compared in Table 3 to a bookkeeping method and a DGVM model ensemble average from the Global Carbon Project (GCP, Le Quéré et al. (2016)). $E_{\text{LUC}}$ in the net land-use configuration $M_{net,net}$ is considerably smaller than the estimates of the GCP with an average annual emissions of 0.64 PgC yr$^{-1}$ between 1960 and 2009, compared
to the 1.4 PgC yr$^{-1}$ of the bookkeeping approach and the 1.2 PgC yr$^{-1}$ of the multi-model DGVM approach. The emissions of the gross land-use configuration with gross skill scores are higher but still fairly low with 0.88 PgC yr$^{-1}$. The version featuring gross land-use with net skill scores yields higher land-use emissions with 1.07 PgC yr$^{-1}$, which is within the uncertainties of both estimates. The largest discrepancy between LPX and GCP is found in the 1990s and 2000s. The uncertainty in the parameter ensembles is comparable to the uncertainty in the multi-model ensemble of the GCP. The tendency to low emissions
is a consequence of the ensemble favoring low emissions to match the observational total land-atmosphere flux, combined with a relatively weak residual terrestrial sink in LPX-Bern.





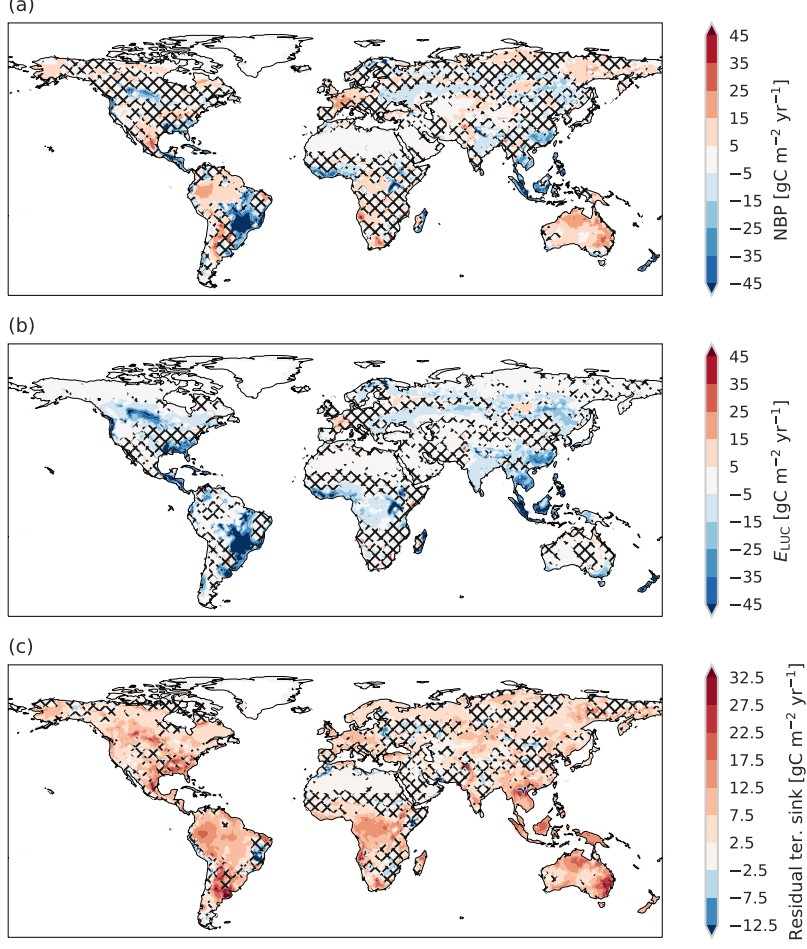

**Figure 4.** Skill weighted median annual net biome production (NBP) (a), atmosphere-land flux due to land-use change $E_{\mathrm{LUC}}$ (b) and the residual terrestrial sink flux (c) from 1901-2016 for the ensemble $\mathrm{M}_{gross,net}$. Areas, where the lower and upper limit of the 90% confidence interval have different signs, are hatched.

In the following the ensemble version with gross land-use and skill scores from the net land-use ensemble $\mathrm{M}_{gross,net}$ is used to investigate the spatial structure of $E_{\mathrm{LUC}}$. This is motivated by the much better representation of the vegetation carbon benchmark in the $\mathrm{M}_{gross,net}$ ensemble than in the $\mathrm{M}_{gross,gross}$ and a higher confidence in the overall benchmark performance of the $\mathrm{M}_{net,net}$ ensemble. A caveat of this choice is that the net land-atmosphere flux is underestimated. However, this should
5   not influence our conclusions on $E_{\mathrm{LUC}}$.





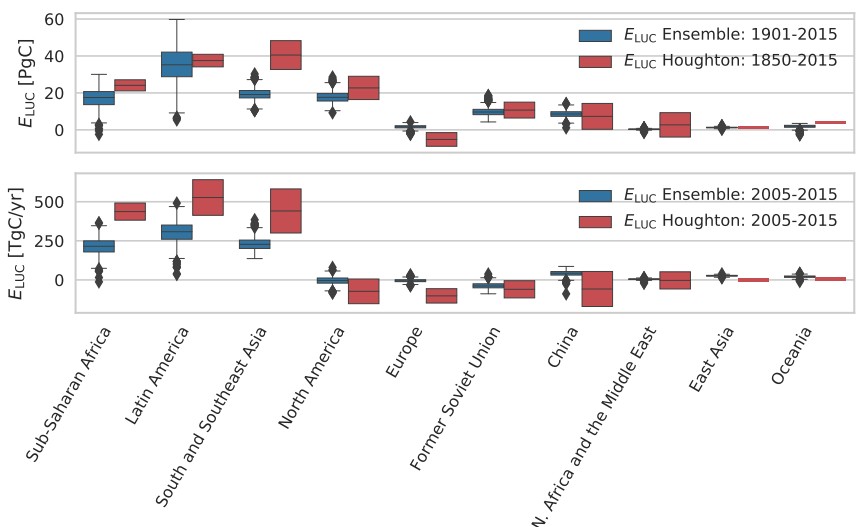

**Figure 5.** $E_{\mathrm{LUC}}$ as simulated by the ensemble $\mathrm{M}_{gross,net}$ compared to the Houghton and Nassikas (2017) estimates. The $E_{\mathrm{LUC}}$ of the ensemble was calculated for individual countries and then aggregated to 10 regions defined in Houghton and Nassikas (2017). The top panel shows the estimates for the emissions over the industrial period; for LPX gridded output is only available after 1901. The bottom panel shows the annual mean emissions from 2005 to 2015. The Houghton and Nassikas (2017) estimates include the reported uncertainties based on the standard deviation of five quasi-independent studies; for East Asia and Oceania no uncertainty is reported.

### 3.1.2 Spatial Patterns and Regional Aggregates

The land-atmosphere fluxes show large regional differences (Fig. 4). The most pronounced feature of net atmosphere-land fluxes is the release of carbon due to deforestation in the Amazon rainforest and the regions close to the equator and a tendency to a net uptake of carbon at higher latitudes, such as central Europe. The calculated land-use change flux is negative everywhere

except central Europe and the west coast of Northern America, resulting in the expected overall emission of carbon due to land-use change. The residual carbon uptake, that is the total atmosphere-land flux minus the contribution of land-use change, shows a consistent uptake of carbon between 1901-2016, with the exception of some areas with high ensemble uncertainty. There are large regions where the 90% confidence interval in the ensemble does not agree on the sign, however, most of these areas feature low NBP.

The $E_{\mathrm{LUC}}$ of $\mathrm{M}_{gross,net}$ are aggregated to regions and compared to estimates of Houghton and Nassikas (2017) (Fig. 5). Since spatial output in LPX is only available after 1901 in LPX, the period 1850 to 2015 in Houghton and Nassikas (2017) is approximated by the interval 1901 to 2015. The global skill weighted median $E_{\mathrm{LUC}}$ from 1850 to 1900 amounts to 24.5 PgC. Overall the global median emissions between 1850 and 2015 in LPX amount to 144.5 (97.5, 192.7) PgC very close to the estimate Houghton and Nassikas (2017) of 145.5 ± 16.0 PgC. The largest discrepancy in the individual regions is found in

South and Southeast Asia, where LPX yields lower emission estimates, which might be a consequence of the lack of tropical





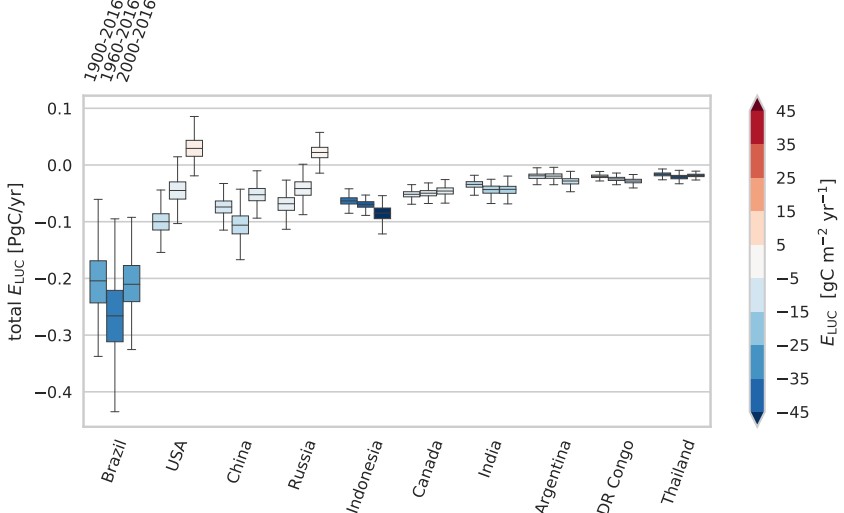

**Figure 6.** Overview of 10 countries with the highest overall contribution to emissions due to land-use change from 1901-2016 in the model ensemble $M_{gross,net}$. The three columns of the bar plot of each country show the total land-use change flux from 1900-2016, 1960-2016 and 2000-2016 respectively. The color of the bar plot corresponds to the land-use change flux per unit area from the respective country and period.

peatlands in the ensemble. In the recent decade from 2005 to 2015, the agreement is less pronounced. While the global annual flux simulated by LPX of 866 (552, 1181) TgC yr$^{-1}$ is within the uncertainty of the independent estimate of $1113 \pm 345$ TgC yr$^{-1}$, the distribution of this flux to the regions shows some divergence. In LPX the tropical regions yield lower emissions, which is somewhat offset by a weaker sink effect in the temperate regions of North America, Europe, China and the former Soviet Union.

By using the NaturalEarthData administrative borders the $E_{LUC}$ of $M_{gross,net}$ are aggregated to individual countries. The $E_{LUC}$ of the ten countries with the largest contribution to total $E_{LUC}$ from 1901-2016 are shown in Fig. 6. Brazil emitted the most carbon due to land-use change, because of the size of the country combined with the high emissions per unit area. The United States of America, China and Russia have moderate per unit area emissions but are a large contributor due to their sheer size. These 3 countries show a decrease of emissions in the 21st century, with the USA and Russia having negative emissions in the 2000s. Indonesia shows the largest per area emissions of the considered countries and emissions increase in the 2000s. The emissions in Indonesia are likely underestimated due to a lack of tropical peatlands in the ensemble.

## 3.2 Evaluation of ensemble performance with respect to observational targets

In this section, the performance of the net land-use ensemble members ($M_{net,net}$; $M_{gross,net}$ performance is nearly identical) in the different observational metrics are discussed. In Fig. 7 a mapping of the $MSE_{rel}$ to an individual skill score is displayed





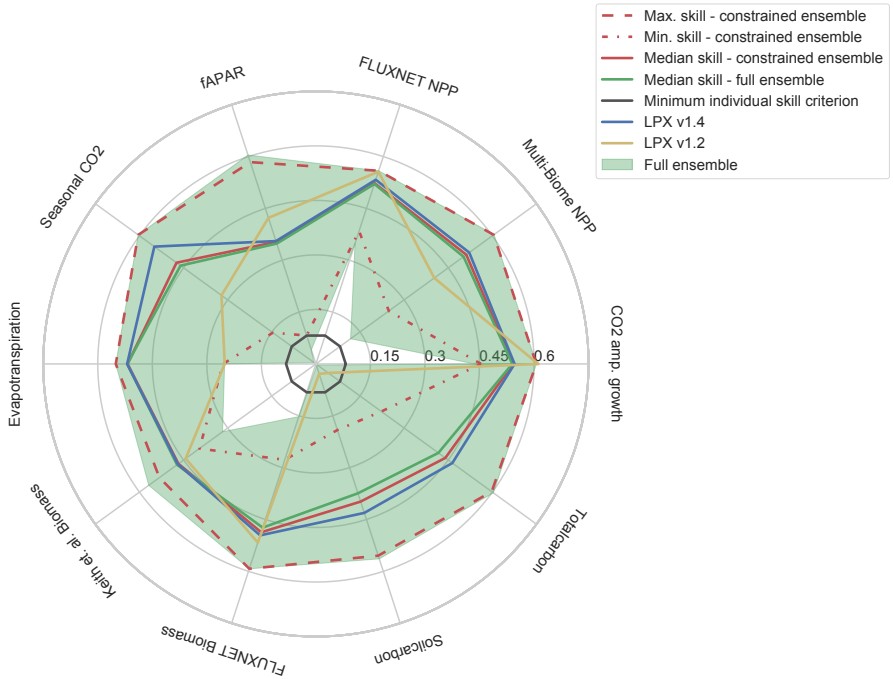

**Figure 7.** A mapping of the $MSE_{rel}$ of observational targets with spatial structure to an individual skill $s = e^{-\frac{1}{2} MSE_{rel}}$ for the $M_{net,net}$ ensemble. The range of the unconstrained ensemble is shaded in green. The range and median of the constrained ensemble is outlined in red. The skill of the new model reference version 1.4 (blue) is compared to the skill of the older model version 1.2 (red). The minimum $MSE_{rel}$ criterion is shown in black.

for the observational data-sets with a spatial structure. With the exception of the biomass measurements by (Keith et al., 2009) and the fAPAR benchmark, the maximum skill in the constrained ensemble is identical to the full ensemble. The reduced maximum skill in those benchmarks is due to an exclusion of singular runs excelling at this benchmark but performing badly in others. The median skill is as expected better or equal for the constrained ensemble than for the full ensemble. The minimum

5 skill is significantly enhanced in the constrained ensemble. In all but the fAPAR benchmark the skill is consistently higher than the minimum skill criterion. LPX v1.4, indicative of the $M_{net,net}$ ensemble performance, is compared to the observational targets in more detail in the supplementary Figures S1-S14.

As an illustration of the observational constraints, we consider the seasonal cycle of atmospheric $CO_2$ and the growth in the amplitude of the seasonal cycle of atmospheric $CO_2$. In Fig. 8 the median simulated values, as well as the 90% confidence

10 interval, of the $M_{net,net}$ ensemble are compared to the atmospheric measurements (GLOBALVIEW-CO2, 2013) for a subset of 2 measurement sites, Alert (Nunavut, Canada) and Terceira Island (Azores, Portugal). The model ensemble is able to reproduce the seasonality pattern, as well as the increase in seasonal amplitude.

For the scalar targets, the median values and range of the full and constrained ensemble are compared in Fig. 9. The constrained ensemble shows a consistently improved performance for the uptake targets. In general, the targets are matched well





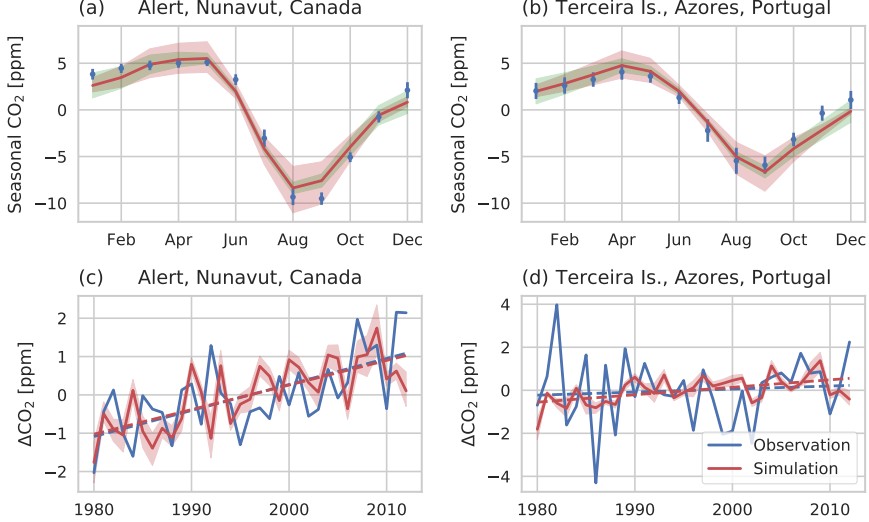

**Figure 8.** Panels (a) and (b): Seasonal cycle averaged from 1980 to 2013 at 2 measurement sites (GLOBALVIEW-CO2, 2013) (blue) compared to the median values of the $M_{net,net}$ ensemble, with 90% confidence interval shaded in red. The standard deviations of the seasonal average are indicated with error bars for the measurements and green shading for the simulations. In panels (c) and (d) the growth in the amplitude of atmospheric $CO_2$ for the same 2 measurement sites (GLOBALVIEW-CO2, 2013) (blue) are compared to the median of the model ensemble, with the 90% confidence interval shaded in red. A linear fit indicated by dashed lines is included. The $CO_2$ concentration at a given site and time is computed with the TM2 transport model using simulated net land-atmosphere fluxes for each ensemble member and ocean-atmosphere fluxes from the Bern3D ocean model (Battaglia and Joos, 2017). The seasonal cycle of $CO_2$ is dominated by fluxes from the land, in particular, the northern hemisphere.

for the 20th century but net land carbon uptake is underestimated in the model ensemble compared to the observational estimates in the beginning of the 21st century. Soil carbon and vegetation carbon inventory are matched well in the model, with a considerable decrease of model spread in the constrained ensemble. The median vegetation carbon of the constrained ensemble is lower than the full ensemble. This is due to a trade-off in the skill of land carbon uptake, increased vegetation carbon leads
5 to a higher release of carbon due to deforestation.

Vegetation carbon inventory and spatial distribution are highly significant for $E_{LUC}$ estimates (Li et al., 2017). The sum of the vegetation carbon estimate and soil carbon estimate by Carvalhais et al. (2014) is used as a constraint for the total carbon, however, the individual vegetation carbon data is not used as a constraint. Nevertheless, the global vegetation carbon inventories of the two products are compatible with 422 (328,523) PgC for the vegetation carbon as simulated by LPX and 449
10 PgC for the Carvalhais et al. (2014) estimate. The spatial patterns (Fig. 10) between simulated vegetation and the Carvalhais et al. (2014) estimates are fairly consistent with a correlation between the two products of $r^2 = 0.83$. LPX simulates somewhat more carbon in vegetation in the high latitude. The extent of areas with high vegetation density in tropical Africa is larger in





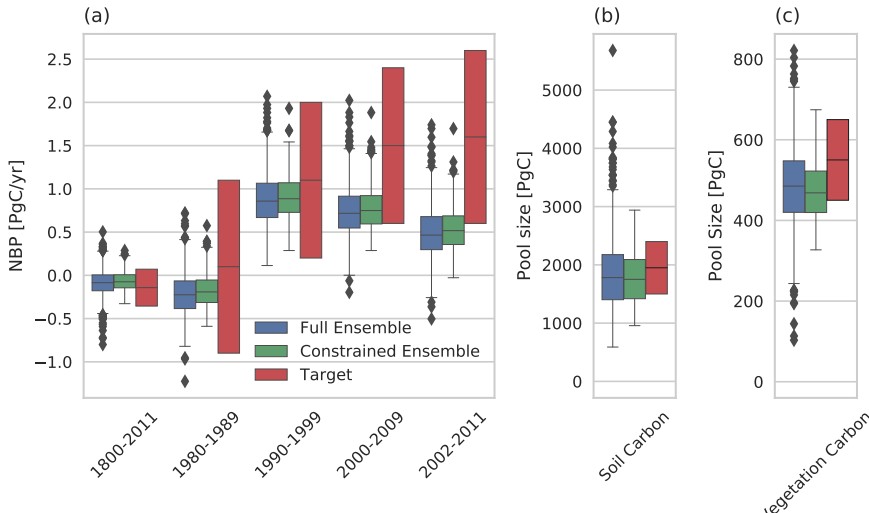

**Figure 9.** The value and uncertainty of the scalar targets (red) compared to an unweighted histogram of the full (blue) and constrained (green) ensemble $M_{net,net}$. Panel (a) shows the net biome production in 5 periods, panels (b) and (c) show the global soil and vegetation carbon inventories respectively.

LPX, but peak vegetation density in this area is lower than in the observational product. The vegetation carbon density in the model is somewhat lower in south-east Asia.

We compare the total land-atmosphere exchange flux to an inversion using anthropogenic fossil fuel emissions (Boden et al., 2017), atmospheric $CO_2$ concentration (Rubino et al., 2013) and an ensemble simulation of the Bern3D ocean model in Fig.

11. The model ensemble shows lower emissions in the early 20th-century and slightly underestimates NBP in the latter half of the 20th-century compared to the inversion. The overall exchange of carbon over the industrial period is within the uncertainty of the estimate.

The unweighted kernel density estimates of the prior (full ensemble) and posterior (constrained ensemble) parameter distributions are shown in Fig. 1. The iterative procedure discussed in 2.3 results in only slight changes of the posterior distribution

with respect to the prior distribution. The median of the distributions is however substantially different from the initial parameter value used in LPX v1.2, the version used as a starting point for this study.

### 3.3 Parameters of the new reference model version

We use the constrained ensemble to establish a new reference model version, featuring a set of optimized parameters. The reference version will be used for model simulations where the use of an ensemble is not appropriate or required.

The skill weighted median parameter values of the constrained ensemble are used as a reference model and its parameter values are shown in Table 1. In Fig. 11 cumulative NBP is displayed for an older model version, the mean values of constrained





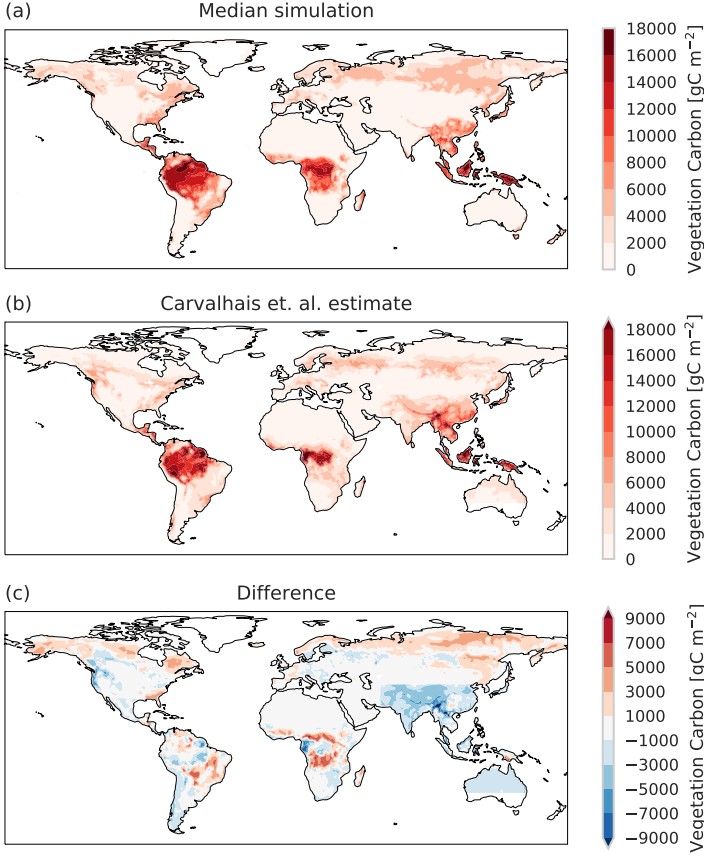

**Figure 10.** The skill weighted median $M_{net,net}$ vegetation carbon distribution averaged between 1982 and 2005 (a), compared to the Carvalhais et al. (2014) vegetation carbon estimate (b). The correlation of the estimates is $r^2 = 0.83$. The absolute difference is shown in panel (c).

and unconstrained model ensemble as well as a run with the new best guess parameters. The best guess version is similar to the mean behavior of the constrained ensemble, showing a net uptake of carbon in the latter half of the $20^{th}$ century, consistent with observations (Ciais et al., 2013). We note that the intermediate version v1.3 used in Keller et al. 2017 features similar parameter settings as determined here. This version simulated 20th century changes in carbon isotope discrimination and intrinsic water use efficiency in good agreement with tree-ring data. The severe underestimation of the land-carbon sink in older versions of LPX-Bern was a consequence of the introduction of new features and improvements in the code of LPX-



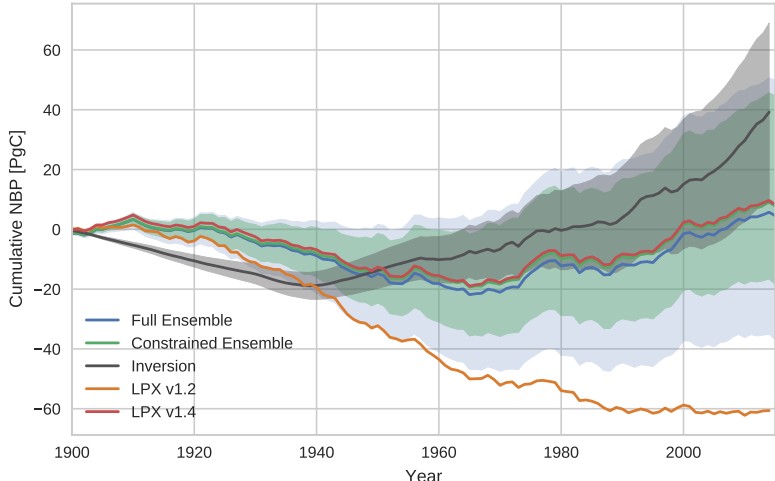

**Figure 11.** Cumultative net biome production (NBP) of the unconstrained (blue) and constrained (green) ensemble with 90% confidence interval shaded, LPX Version 1.2 (orange) and the new reference model version 1.4 (red). The result of a so-called "single deconvolution" is shown by the black line and grey range. In this inversion, the change in the land inventory is inferred from the records of atmospheric $CO_2$ and anthropogenic emissions from fossil fuel (and cement production) and ocean carbon uptake as estimated from an ensemble of simulations with the Bern3D model.

Bern, without subsequent retuning of the parametrization. The parameter changes are most pronounced in the temperature dependence of heterotrophic respiration $E_{0,hr}$ and $\alpha_m$, a parameter associated with plant water demand. Both of these changes are not unexpected, as they increase the land carbon sink. In the case of heterotrophic respiration less carbon is lost due to increasing surface temperature and the increased water demand amplifies the $CO_2$ fertilization effect.

5    Overall the updated parametrization shows a well-balanced performance in the spatial benchmarks shown in Fig. 7. The older LPX version excels at singular metrics, namely the amplitude growth of $CO_2$ and the FLUXNET measurements, but breaks down at others, such as the spatial distribution of carbon and evapotranspiration. Furthermore, it also performs considerably worse in the scalar and inversion targets.

The choice of using the skill weighted median parameters of the constrained ensemble instead of simply using the best

10    performing parameter set for the reference version is motivated by its robustness and representativeness of the ensemble. While the best performing model member certainly possesses a higher skill score, its parameter values can depend strongly on the choice and weighting of the observational targets, whereas the median parameter values depend less on individual metrics.





## 4 Discussion

### 4.1 Land-atmosphere fluxes and $E_{\mathrm{LUC}}$

The simultaneous assimilation of multiple observational constraints yields soil and vegetation stocks and distributions which are consistent with observations. The total land-atmosphere carbon flux is reproduced relatively well in the model configuration

using net land-use $M_{net,net}$. Comparing the land-atmosphere carbon flux to the independent flux estimates by Schimel et al. (2015) in the period 1990-2007, the tropical and southern fluxes are in good agreement to the atmospheric inversion results with airborne constraint with a flux of 0.24 (-0.02,0.57) in LPX-Bern. The flux in the northern extratropical areas of 0.50 (0.37,0.63) is on the lower end but easily fulfills the mass balance.

    The observed uncertainties of $E_{\mathrm{LUC}}$ due to parameter uncertainty in the DGVM LPX is on the same order of magnitude as

structural uncertainties, such as including or not including modules for shifting cultivation and wood harvest. The effect of the inclusion of additional land-use processes can even be compensated by a change of parametrization, while still conforming to the observational benchmarks, indicating that it might be possible to capture the magnitude of $E_{\mathrm{LUC}}$, while neglecting second order processes. The compensation of $E_{\mathrm{LUC}}$ occurs because the residual sink is less sensitive to parametrization changes than the $E_{\mathrm{LUC}}$ in LPX-Bern. This behavior has also lead to an $E_{\mathrm{LUC}}$ that is on the lower end of independent estimates (Le Quéré

et al., 2016). A lack of large difference in $E_{\mathrm{LUC}}$ from model setups featuring gross and net land-use might seem in contrast with the result of other studies investigating these processes (Arneth et al., 2017; Wilkenskjeld et al., 2014; Stocker et al., 2014; Shevliakova et al., 2009), however, if we keep parametrization constant ($M_{gross,net}$) we find the expected lower $E_{\mathrm{LUC}}$ for net land-use.

    We investigated the magnitude and spatial distribution of $E_{\mathrm{LUC}}$ in the model configuration using the skill scores and

parametrization from the standard net land-use configuration with additional processes of shifting cultivation and wood harvest ($M_{gross,net}$). This choice is motivated by the good performance of the net land-use ensemble in the observational benchmarks (Section 3.2 and Figures S1-S14).

    A good correspondence between simulated fluxes and the estimates of Houghton and Nassikas (2017) in 10 regions during the industrial period is found. When comparing recent decades, LPX-Bern generally seems to simulate lower $E_{\mathrm{LUC}}$ than both

the bookkeeping approach based estimate and the aggregated estimates in the GCP. The biggest disparity is comparatively low fluxes in the South and Southeast Asian regions in LPX-Bern, which are at least partially explained by the lack of tropical peatlands in this model configuration. The burning and draining of tropical peatlands is an important contribution to $E_{\mathrm{LUC}}$ in tropical regions (Maria Roman-Cuesta et al., 2016; Koh et al., 2011; Hooijer et al., 2010). The annual emissions estimate from draining peatlands used in Houghton and Nassikas (2017), increase from almost no emissions in 1980 to roughly 0.2

PgC yr$^{-1}$ in 2015. The lack of tropical peatlands is also consistent with the underestimated soil carbon density in these regions when compared to Carvalhais et al. (2014). Other studies suggest higher historical $E_{\mathrm{LUC}}$, such as the bookkeeping approach by Hansis et al. (2015), including shifting cultivation, with an estimate of 261 PgC between 1850-2005. Some of the difference between DGVM model results and bookkeeping approaches can be attributed to different definitions of LULCC emission (Pongratz et al., 2014; Stocker and Joos, 2015).





A recent study by Li et al. (2017) constrained $E_{\mathrm{LUC}}$ by using biomass observations. They derived a relationship between $E_{\mathrm{LUC}}$ and biomass in nine regions using the nine DGVMs in the TRENDY-v2 model intercomparison (Sitch et al., 2015) and applied empirical estimates for biomass carbon to arrive at a constrained $E_{\mathrm{LUC}}$ of 155 ± 50 PgC between 1901 and 2012. The result of 116 (77, 156) PgC as in this study is compatible, albeit somewhat lower. By neglecting all other constraints and exclusively using the global vegetation carbon by IPCC (Ciais et al., 2013) and the biomass map by (Carvalhais et al., 2014) (Also used as one of the constraints in Li et al. (2017)) as constraints, we arrive at a higher $E_{\mathrm{LUC}}$ of 130 (87, 179) PgC. This illustrates the importance of the biomass inventory for the magnitude of $E_{\mathrm{LUC}}$.

$E_{\mathrm{LUC}}$ is not only influenced by uncertain model processes and parametrizations but also the underlying LULUC forcings (Goll et al., 2015). Peng et al. (2017) have shown that the choice of transition rules, governing how new land-use areas are allocated from previous areas, has a considerable effect on $E_{\mathrm{LUC}}$. The effects of these uncertainties are not accounted for in this study since we only use one land-cover forcing product and one set of transition rules is used.

Overall the ensemble approach produces $E_{\mathrm{LUC}}$ estimates consistent with other independent estimates, albeit somewhat on the lower end of the range of estimates. This is a consequence of the constraining process favoring parametrization with low $E_{\mathrm{LUC}}$ over a high residual sink, which is discussed further in the next section.

## 4.2 Benchmark performance and best guess version

A hierarchical weighting scheme to compare a diverse set of constraints was employed, following earlier work (Steinacher et al., 2013). A set of 14 data sets (Fig. 2, Table 2) was selected to constrain the model's performance with regard to steady state carbon and water fluxes and carbon inventories as well as with regard to transient changes. Globally aggregated as well as spatially resolved information is used to constrain simulated spatial patterns and to robustly model global mean properties. The temporal focus is on the decadal-to-century time scales most relevant for projections of anthropogenic climate-carbon cycle changes and on the seasonal cycle of photosynthesis and the decadal amplification of the seasonal cycle in land-atmosphere fluxes (McGuire et al., 2001; Graven et al., 2013) which provide information on underlying processes. The iterative procedure for choosing the prior parameter distribution yielded an ensemble which performs well with respect to the selected metrics.

In addition to the weighting of model results with the global skill score, we employed a minimum skill criterion, discarding runs with very bad performance in a singular metric. This approach is somewhat comparable to pre-calibration methods, where implausible parameter spaces are also ruled out (Williamson et al., 2017; Holden et al., 2010; Edwards et al., 2011), and aims to sensibly reduce the size of the parameter space.

While the uptake of carbon by the terrestrial biosphere in the model ensemble is significantly larger than earlier versions of LPX, it is still in the lower range of estimates. A direct way of increasing the magnitude of change in land carbon is to change pool sizes, which is here restricted by other observational constraints. The inclusion of more processes, such as natural and human-induced erosion (Wang et al., 2017) could also increase the strength of the terrestrial sink, however other processes such as shifting cultivation lead to a decrease of the land carbon sink. A further possibility is the revising of established processes in the model. The climatic dependence of the auto- and heterotrophic respiration is an important component, mitigating the



$CO_2$ fertilization effect. The implementation of a more refined module might decrease this negative feedback, thus increasing carbon storage and sink sensitivity.

The sink strength could potentially also be enhanced by including so far not included parameters and including additional constraints that discriminate between the different components of the land sink.

The release of both spatially and temporally resolved carbon flux observations by using remote sensing, such as the Carbon Monitoring System Flux Pilot (CMS) project, featuring not only net fluxes but also gross production and respiration, is a very promising candidate for constraining the parameter space further. The spatial structure might restrict the apparent degree of freedom in partitioning the terrestrial sink in $E_{\mathrm{LUC}}$ and residual land carbon sink. $\delta^{13}$C isotope measurements in vegetation also have the potential to be a useful additional constraint in land biosphere models (Keller et al., 2017).

The simultaneous assimilation of multiple observational constraints allowed to formulate a well rounded best guess version of the model. While this parameter version doesn't necessarily excel at every single benchmark, it shows a consistent performance amongst all different targets. This behavior leads us to believe that the best-guess version is well suited for simulations spanning long time spans, both for paleo and future research questions, where the use of a full parameter ensemble is not feasible. Furthermore, it can also be used in model intercomparison studies, where single realizations of different models are

compared.

## 5   Conclusions

We successfully applied a multi-purpose model benchmark to a perturbed parameter ensemble, obtained with a Monte-Carlo like sampling technique, of a dynamic global vegetation model (DGVM). Specifically, we developed a "best-guess" model version and constrained the residual carbon sink flux and carbon emissions from anthropogenic land-use ($E_{\mathrm{LUC}}$) over the

industrial period. The general characteristics of the framework are as follow. (i) The framework permits a standardized model benchmarking (Hoffman et al., 2017; Kelley et al., 2013; Luo et al., 2012; Blyth et al., 2011) by comparing different models or model versions graphically and using statistical metrics (Stow et al., 2009) to a broad and diverse range of observations. (ii) The efficient Latin Hypercube sampling method (McKay et al., 1979) is used to explore the model parameter space and to set up and run perturbed parameter ensembles for a large set of model parameters. The advantage of the Latin Hypercube

sampling is the representative sampling of different parameter combinations, whereas a shortcoming is that the sampling size has to be determined in advance. (iii) A hierarchical model weighting scheme is used to assimilate diverse observations. These may differ with respect to spatial and temporal resolution and quality and include observations from the local scale, such as data from individual biomass measurements or the seasonal $CO_2$ cycle at individual atmospheric sampling sites, up to global scale gridded data products such as satellite measurements of absorbed radiation by plants. A major advantage of this scheme

compared to sequential assimilation techniques such as Ensemble Kalman Filters is that the influence of necessarily subjective choices (Rougier, 2007) on the results can be investigated a posteriori; in other words without performing costly additional simulations. The subjective choices may be of scientific nature such as whether an observational data set is considered or not or of more technical nature such as whether gridded data values are weighted by grid cell area or not. (iv) The applied modular



framework is easily extendable to incorporate different or more observational constraints and to different mechanistic models including other DGVMs, ocean models (Battaglia et al., 2016) or Earth System Models (Steinacher et al., 2013; Steinacher and Joos, 2016)). (v) The Bayesian, skill-score weighted ensemble is able to constrain the median and uncertainty ranges of unknown or uncertain quantities such as carbon emissions from anthropogenic land-use, marine nitrous oxide production

(Battaglia and Joos, 2017), or climate sensitivity metrics (Steinacher and Joos, 2016) (vi) Finally, the skill-score weighted ensemble is suitable for probabilistic projections including both likely and less likely model configurations and assumptions.

A new reference version of the LPX-Bern (v1.4) DGVM was established. We were able to show that the constrained ensemble, as well as a resulting best guess version, perform consistently well under a range of benchmarks (Table 2) while satisfying a minimum skill criterion in every single benchmark. The best guess version was formulated using the weighted median param-

eter values of the constrained ensemble, instead of using the parametrization of the overall best run. This choice is motivated by the robustness of the resulting parameter values with respect to changes in the hierarchical weighting scheme and the in- or exclusion of individual observational targets and its representativeness of the perturbed parameter ensemble. The new model version LPX-Bern v1.4 successfully simulates observation-based estimates of the cumulative net land uptake and release over the industrial period.

Many previous studies have investigated inherent uncertainties in ELUC estimates (Houghton et al., 2012; Goll et al., 2015; Peng et al., 2017). Our study aims to contribute to this ongoing discussion by providing DGVM $E_{\mathrm{LUC}}$ uncertainty estimates purely due to parameter uncertainty in an observationally constrained model ensemble using the LUH2 v2h (Hurtt et al.) product. Overall the benchmarking scheme favors runs with low emissions due to a relatively low residual sink sensitivity in the model and constraining total land-atmosphere fluxes. We consider model ensembles with and without additional land-use

processes (shifting cultivation and wood harvest) and find that the difference in global $E_{\mathrm{LUC}}$ is on the same order of magnitude as parameter induced uncertainty. The inclusion of shifting cultivation and wood harvesting increases emissions similar in magnitude to earlier studies (Stocker et al., 2014; Shevliakova et al., 2009) when applying the same model parameters, while in some cases these additional emissions could potentially even be offset with appropriate parameter choice. We attributed the fluxes to different countries and closer investigated the ten countries with the most emissions in the industrial period due to

land-use and land-use change. Our land-use carbon emission estimates are similar to those of Houghton and Nassikas (2017) on the country level and overall consistent with other independent estimates on regional to global levels (Li et al., 2017; Le Quéré et al., 2016).

The observation-constrained DGVM ensemble and best guess version established in this work are ready for use in model intercomparison studies (Tian et al., 2018; Sitch et al., 2015) and longer time span paleo simulations. It may also be applied to

quantify future terrestrial carbon fluxes and $E_{\mathrm{LUC}}$ for different shared socio-economic pathways. Additional new observational data streams may be implemented in our modular framework to further refine results.

*Data availability.* Model output is available upon request to the corresponding author (lienert@climate.unibe.ch).



*Competing interests.* The authors declare that they have no conflict of interest.

*Acknowledgements.* We thank G. Battaglia for supplying the Bern3D model output and M. Scholze for providing the TM2 transport matrices. We would like to thank the data community for their efforts in providing high quality data sets. This work was supported by the Swiss National Science Foundation (#200020_172476).



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
