# Peer review of "A Bayesian Ensemble Data Assimilation to Constrain Model Parameters and Land Use Carbon Emissions"

_Biogeosciences, 2018_

## Referee Comment (RC1) · S. Zaehle (Referee) · 1 Mar 2018

Lienert and Joos apply a bayesian data assimilation framework to the LPX-Bern model in order to constrain a selection of model parameters using a range of local to global carbon and water cycle observations. In the manuscript, they describe the framework and illustrate the key model performance criteria. This framework allows them to provide a data-constrained simulation of the regional and global terrestrial carbon balance between 1860 and 2016, and in particular to estimate the land-use related carbon emission, including an uncertainty range.

This is a very good study integrating multiple observations in a systematic and repro-

ducable way to constrain a process-based global carbon cycle model. This system is not only used to produce a newly calibrated LPX-Bern version for future use, but also to provide useful insight into the magnitude (and particularly the uncertainty) of land-use emissions. Overall this is a valid contribution to Biogeosciences.

Unfortunately, I have troubles following the method description. The description of the way, the parameter distributions are updated is remains fairly unclear. I recommend that the authors devote a special section in the Methods section to clarify a couple of points: a) how was the prior distribution of the parameters derived (literature ranges typically only allow to assume uniform distributions); b) how exactly is the ensemble updated after the metrics are calculated. Is it the probability distribution of each parameter, which is updates? This would lead to a new LHS set to be produced, and subsequent new model runs? Figure 1 would suggest that this is the case, but in this case, the new set would be dependent on the metric and metric weighting, which contradicts the statements made on P2 (also, it's computationally probably prohibitive). Or is each LHS sample weighted according to the model performance, and this weight then used to calculate the PDF of a modelled output? If that is the case, I don't understand the iterative nature of the LHS sample updates? Also, in this case, it would be good if the authors would elaborate on the way they've estimates the posteriori distributions. Given that the authors highlight the ability to change the cost-function and weighting as a key strength of their method, it would be also interesting if they would add a discussion point as to how robust they believe the posteriori parameter distributions are against their choice of metric & weighting.

I have a number of further suggestions to improve the clarity of the manuscript:

P1 L17: in the context of a data assimilation paper, the use of assimilated here is confusing. replace by stored?

P2 L3: Add "Amongst others," at the beginning of the sentence

P2 L4: unclear what uncertain prescribed LULCC processes are meant to be, perhaps

give examples, or clarify that it's the representation of these processes that is uncertain

P2 L9 DA "should" be an integral part of model development, but unfortunately it is not always.

P2 L10: Is the Houwelling reference appropriate here? This does mostly relate to inverse atmospheric modelling

P2 L12: Not sure that I understand sequentially correct here. Most DA methods would assimilate different data sources simulateneously. Also, I think cost-function is the more common term for metric in this context

P2 L14 This sentence is a bit out of context in a paragraph on alternative DA methods, because benchmarking does in general not imply DA. It seems more logical to merge this sentence with the Paragraph starting in L25, and move the entire paragraph to L6 after Le Quere et al. 2016.

L2 L18: As noted above, I have troubles following here: LHS simply provides a set of parameter combinations, in which each parameter is sampled given a specified distribution and notably, ensuring that there is no correlation amongst any of the parameters. LHS does not imply any model metric per se. The way the posterior distribution is derived from the prior distribution and the model metrics is unclear. How many iterations would be needed to arrive at a stable solution, what is the stopping criteria, and why is it possible to change the metric during the DA procedure? This would change the posterior distribution, and therefore impede convergence.

P2 L31: I wonder if the flow of the introduction would be more logical if one would first talk about the LULCC processes as in this paragraph, then about the benchmarking in the preceeding paragraph, and only then about data assimilation?

P2 L31: While the (add) "net" land-atmosphere flux can "to some extent" be . . .

P2 L32: add "residual" terrestrial carbon sink?

P4 L21: I think it is worth highlighting that the strength of LHS over other sampling techniques is that the set of parameters in uncorrelated.

P4 L20: The text confuses MC parameter sampling techniques, which are indepedent of any purpose the sampling is made for, from MC Data assimilation techniques, which are not?

P4 L26: the description of alpha_a should correspond to table 1, it is not FAPAR!

P4 L7: Literature range only allow to give uniform distribution. How where the non-uniform distribuition parameters obtained / estimated?

P4 L7: I have trouble following from here on. Maybe this would become clearer, if first all the metrics and data sources were explained, and then the way the distributions are updated is clearer presented.

P8 L10: which winds were used for the transport? I assume that the winds were not interannually varying?

L9 L5: Inversion typically refers to the inverse modelling of atmospheric transport, whereas here - as far as I understand this, you simply take the land flux as the residual of the fossil fuel emission and ocean uptake.

P9 L10: Are these data sources not redundant with the global maps of total and soil C storage described earlier?

P9 L27: I don't understand the reasoning for the duplication of ensembles with gross transitions. Please motivate.

P9 L31: As noted above, I have difficulties following this description.

P10 L5-8: Is material for the introduction, not the results section

P10 L 8-11 can be safely removed.

Section 3: When giving numerical estimates, please add either range or standard deviation, whenever the number is based on the ensemble. I also think that the more logical arrangement of the Results sections would be to first talk about model performance, and then about the attribution of the net land flux to LULCC and residual.

P13 L4: Why would an underestimation of the ELUC not affect your conclusions about ELUC?

P17 L6: is the use of the word significant appropriate here?

P17 L8: why not?

P 18 L 3: Why is this different from the approach described in Section 2?

Conclusion Section: There is no need to repeat details of the methods or approach undertaken

Figure 1: Ensure all lines are visible

Table 1: Check units and definition for E0. This seems more like an activation energy to me (not a temperature sensitivity

What are the units of the k_la:sa? Is this simply a scalar?

---

## Referee Comment (RC2) · J.-F. Exbrayat (Referee) · 13 Mar 2018

The study presents an approach to constrain a DGVM with multiple observational streams of carbon stocks, gross and net fluxes. The authors rely on a latin hypercube stratified sampling to perturb model parameters and create several 1,000-member ensemble simulations of the terrestrial carbon cycle for the historical period. Results focus on the estimation of land-use and land-cover change emissions.

This study is quite innovative in the context of the global terrestrial carbon cycle as model parameters are constrained globally. I have found several similarities between the method described here and the Generalised Likelihood Uncertainty Estimation

method used in hydrological sciences (Beven and Binley 1992).

There are several issues with the current manuscript which need to be addressed.

First, my main criticism targets the description of the sampling method. It is very unclear how the prior probability distribution in Figure 1 and the new best-guess values in Table 1 have been obtained, and how the posterior distribution of the parameters is calculated. Is it based on the selection criterion used to exclude the less skilled model parameters (p7 I5-8)? If Figure 1 and Table 1 present results from the current manuscript they should be described in the corresponding section.

Second, I struggle to understand what experiments were actually undertaken. From section 2.6, it seems that three simulations are performed for each parameter set. These three simulations differ in the representation of LULCC: none (reference), gross or net transitions. Then, the results section reports the three different model configurations  $M_{net,net}$ ,  $M_{gross,net}$ ,  $M_{gross,gross}$  while these are first described as three alternative skill weighted median.

Third, I am unclear about the skill-weighted mean method. Simulations with either net or gross land-use configuration are likely to yield different results so it is hard for me to justify  $M_{gross,net}$ . I understand that the  $M_{gross,net}$  skill-weighted mean provides the best results compared to benchmarks (Table 3) but it could be an artefact, couldn't it? Also, some parameter sets are likely to perform better in some regions and worse in other. Therefore, would a spatially-explicit weighting scheme (Schwalm et al., 2015; Exbrayat et al. 2018) be more suited to constrain the ensemble?

Hereafter are some more specific comments

p4 I6: CRU TS3.23 covers 1901-2014, so how are simulations performed for 1800-2014 (or is it 1800-2016 like in the abstract?) please clarify throughout the manuscript

p4 l21: please define what model metrics

p5 I6: how have these distributions been chosen?
p7 I1: please write  $MSE_{rel}^i$

p9 I28: 'LULUC'? please correct here and in several other places

p10 I13-23: please include some information about the uncertainty displayed in the Figures here and throughout the text

p11 I1: please quantify 'slight'

p14 l11: see previous comment on the study period

p16 l3: an informative figure would a covariance matrix of the parameter sets' scores for each criterion

p16 I12: according to Figure 8b and d, the model captures the seasonality but not the interannual variability. This is worth reporting (and explaining).

Fig 1: Mnet,net is not defined

Fig 3: please explain the sign convention as it seems at odd with figure 4 ( $E_{LUC}$  in particular)

Fig 7: this figure is very complicated. Why is it important to look at the whole ensemble, and the constrained one? Constraining the ensemble uncertainty is not a major point in the rest of the manuscript and uncertainties are not reported in most of the text.

Fig 8: please move the legend

References

Beven, K. J. and Binley, A.: The future of distributed models: model calibration and uncertainty prediction, Hydrol. Process., 6, 279–298, 1992.

Exbrayat, J.-F., Bloom, A. A., Falloon, P., Ito, A., Smallman, T. L., and Williams, M.: Reliability ensemble averaging of 21st century projections of terrestrial net primary productivity reduces global and regional uncertainties, Earth Syst. Dynam., 9, 153-165, https://doi.org/10.5194/esd-9-153-2018, 2018.
Schwalm, C. R., Huntzinger, D. N., Fisher, J. B., Michalak, A. M., Bowman, K., Ciais, P., Cook, R., El-Masri, B., Hayes, D., Huang, M., Ito, A., Jain, A., King, A. W., Lei, H., Liu, J., Lu, C., Mao, J., Peng, S., Poulter, B., Ricciuto, D., Schaefer, K., Shi, X., Tao, B., Tian, H., Wang, W., Wei, Y., Yang, J., and Zeng, N.: Toward "optimal" integration of terrestrial biosphere models, Geophys. Res. Lett., 42, 4418–4428, https://doi.org/10.1002/2015GL064002, 2015.

---

## Author Comment (AC1) · 6 Apr 2018

We want to thank the reviewer for the time and effort for the careful and very insightful review. In the following, we respond to the reviewer point by point, with our **responses in bold** and *quotations from the updated manuscript in cursive*. Please also consider the updated manuscript with track changes in the supplementary. Also note that we expanded the discussion section to include a paragraph on a potential bias in the fossil fuel emissions used for the deconvolution by including non-fuel uses.

**Point-by-point response**

The study presents an approach to constrain a DGVM with multiple observational streams of carbon stocks, gross and net fluxes. The authors rely on a latin hypercube stratified sampling to perturb model parameters and create several 1,000-member ensemble simulations of the terrestrial carbon cycle for the historical period. Results focus on the estimation of land-use and land-cover change emissions. This study is quite innovative in the context of the global terrestrial carbon cycle as model parameters are constrained globally.

**Thank you**

Unfortunately, I have troubles following the method description. The description of the way, the parameter distributions are updated is remains fairly unclear. I recommend that the authors devote a special section in the Methods section to clarify a couple of points:

**A separate section on prior selection is added to the manuscript.**

a) how was the prior distribution of the parameters derived (literature ranges typically only allow to assume uniform distributions); b) how exactly is the ensemble updated after the metrics are calculated. Is it the probability distribution of each parameter, which is updates? This would lead to a new LHS set to be produced, and subsequent new model runs? Figure 1 would suggest that this is the case, but in this case, the new set would be dependent on the metric and metric weighting, which contradicts the statements made on P2 (also, it's computationally probably prohibitive). Or is each LHS sample weighted according to the model performance, and this weight then used to calculate the PDF of a modelled output? If that is the case, I don't understand the iterative nature of the LHS sample updates? Also, in this case, it would be good if the authors would elaborate on the way they've estimates the posteriori distributions.

**We introduced a new subsection discussing the method used to obtain the prior parameter distribution and reordered the section for clarity. Please refer to section 2.3.2 (p.6 l.9-p.8 l.7 in the manuscript).**

**Figure 1 in this reply shows the evolution of the median parameter values and ranges of the ensembles with 200 and 300 members (T1-T6) and large ensembles with 1000 members (E1-E3), discussed in the new section in the manuscript. Only the parameters used in the final ensemble E3 are shown. In the small ensembles, different model configuration and parameters were tested. For instance, in ensemble T2 nitrogen limitation was not considered and thus the nitrogen cycle related parameters were not sampled. The factorial simulation and small ensembles informed the choice for the prior of ensemble E1, which is then iteratively improved to arrive at the prior of E3.**

Given that the authors highlight the ability to change the cost-function and weighting as a key strength of their method, it would be also interesting if they would add a discussion point as to how robust they believe the posteriori parameter distributions are against their choice of metric weighting.

**We qualitatively assessed the robustness by reevaluating the ensemble for a subset of the observational targets. For this purpose we created multiple hierarchical weighting schemes, each missing one of the observational targets or a category of targets when compared to the default version, and looked at the induced changes in the parametrization of the best guess version. We added a section discussing this approach to the manuscript:**

*"We investigate the dependency of the constrained ensemble on the choice of the observational constraints by reevaluating the ensemble for a subset of observations. We created 19 weighting schemes, each missing one of the individual observational constraints (Figure 2 and table 2) and otherwise identical to the default scheme. Then the median skill weighted parameter values of these ensembles are compared to the best-guess values of $M_{net,net}$ (section 3.3). The relative change in parameterization is less than 1% for 15 out of the 19 considered alternative weighting schemes. Leaving away the global vegetation and soil carbon constraints lead to moderate changes, notably to a change in the parameter for mortality ($mort_{max}$) of 4% and 2% respectively. Not*

*including the soil carbon distribution in high latitudes lead to an increase of the parameter for the dependency of soil respiration on temperature ($E_{0,hr}$) of 2%. The largest changes in parameterization were observed when not considering the atmospheric deconvolution, most notably the sapwood-heartwood turnover time $\tau_{sapwood}$ decreased by 5%. When omitting entire categories in the benchmarking scheme, the changes in parametrization are larger than for omitting individual constraints, with parameter changes of up to 1% for the fluxes, 5% for the inventory and 6% for the transient category. This shows that the final parameterization is not overly sensitive to the inclusion or omission of a single observational product."*

I have a number of further suggestions to improve the clarity of the manuscript: P1 L17: in the context of a data assimilation paper, the use of assimilated here is confusing. replace by stored?

**Done**

P2 L3: Add "Amongst others," at the beginning of the sentence

**Done**

P2 L4: unclear what uncertain prescribed LULCC processes are meant to be, perhaps give examples, or clarify that it's the representation of these processes that is uncertain

**Done, the sentence now reads:**
*"In addition to uncertainties in the prescribed LULCC forcings and the representation of LULCC and other processes in DGVMs, the values of the applied parameters are subject to substantial uncertainties."*

P2 L9 DA "should" be an integral part of model development, but unfortunately it is not always.

**Replaced "is" by "should be".**

P2 L10: Is the Houwelling reference appropriate here? This does mostly relate to

inverse atmospheric modelling

**Removed Houweling reference**

P2 L12: Not sure that I understand sequentially correct here. Most DA methods would assimilate different data sources simulateneously. Also, I think cost-function is the more common term for metric in this context

**Revised sentence to read:** *"A drawback of these methods is that the sampling process is dependent on the choice of the cost function, the design of which is not trivial when assimilating multiple observations simultaneously."*

P2 L14 This sentence is a bit out of context in a paragraph on alternative DA methods, because benchmarking does in general not imply DA. It seems more logical to merge this sentence with the Paragraph starting in L25, and move the entire paragraph to L6 after Le Quere et al. 2016.

**Moved, the paragraph now reads:**
*"Amongst others, Dynamic Global Vegetation Models (DGVMs) can be used to assess the contribution of LULCC to the terrestrial carbon budget (Le Quéré et al., 2016). The assessment of the performance of a given model version using observational benchmarks has been actively discussed in the literature (Hoffman et al., 2017; Peng et al., 2014; Kelley et al., 2013; Luo et al., 2012; Blyth et al., 2011; Randerson et al., 2009) and different frameworks have been proposed. The selection of observational targets is vital to a successful assimilation of observational data. In order to constrain the contemporary carbon cycle, 14 data products are used, ranging from global inventories of carbon (Ciais et al., 2013) to spatially resolved satellite estimates of photosynthetically absorbed radiation (Gobron et al., 2006). The goal of the data set selection process was to have observations capturing the magnitudes of fluxes and inventories in the carbon cycle, as well as its transient response to anthropogenic perturbance. In addition to uncertainties in the prescribed LULCC forcings and the representation of LULCC and other processes in DGVMs, the values of the applied parameters are*

*subject to substantial uncertainties. We use a Monte-Carlo-like data assimilation approach (Steinacher et al., 2013; Steinacher and Joos, 2016; Battaglia and Joos, 2017) to sample 15 key model parameters and construct a 1000-member model ensemble to investigate this parameter related uncertainty in the DGVM LPX-Bern. Furthermore, we establish a new reference version of the model."*

L2 L18: As noted above, I have troubles following here: LHS simply provides a set of parameter combinations, in which each parameter is sampled given a specified distribution and notably, ensuring that there is no correlation amongst any of the parameters. LHS does not imply any model metric per se. The way the posterior distribution is derived from the prior distribution and the model metrics is unclear. How many iterations would be needed to arrive at a stable solution, what is the stopping criteria, and why is it possible to change the metric during the DA procedure? This would change the posterior distribution, and therefore impede convergence.

**Please see the answer to the major points.**

P2 L31: I wonder if the flow of the introduction would be more logical if one would first talk about the LULCC processes as in this paragraph, then about the benchmarking in the preceding paragraph, and only then about data assimilation?

**We have restructured the introduction but slightly deviated from the reviewers suggestion to improve text flow. Please also see the attached manuscript with track changes.**

P2 L31: While the (add) "net" land-atmosphere flux can "to some extent" be . . .

**Done**

P2 L32: add "residual" terrestrial carbon sink?

**Done**

P4 L21: I think it is worth highlighting that the strength of LHS over other sampling

techniques is that the set of parameters in uncorrelated.

**Done:**

*"..to generate an uncorrelated parameter ensemble of a given size."*

P4 L20: The text confuses MC parameter sampling techniques, which are indepedent of any purpose the sampling is made for, from MC Data assimilation techniques, which are not?

**Changed "Monte Carlo sampling techniques" to "Monte Carlo data assimilation techniques"**

P4 L26: the description of alpha_a should correspond to table 1, it is not FAPAR!

**Done:**

*"The fraction of photosynthetically active radiation assimilitated at ecosystem level relative to leaf level, $\alpha_a$ .."*

P4 L7: Literature range only allow to give uniform distribution. How where the nonuniform distribuition parameters obtained / estimated?

**See answer to major points and attached manuscript.**

P4 L7: I have trouble following from here on. Maybe this would become clearer, if first all the metrics and data sources were explained, and then the way the distributions are updated is clearer presented.

**Revised this section (See answer to major points)**

P8 L10: which winds were used for the transport? I assume that the winds were not interannually varying?

**Yes, the transport matrix does not include interannual variability. Added:**
*This method does not include the interannual variability of the transport.* **Additionally, we added an explanation to Figure 8 for clarity:**

*"As expected, the interannual variability in seasonal amplitude of $CO_2$ is not captured as the atmospheric transport model TM2 does not represent interannual variability in mass transport."*

L9 L5: Inversion typically refers to the inverse modeling of atmospheric transport, whereas here - as far as I understand this, you simply take the land flux as the residual of the fossil fuel emission and ocean uptake.

**We changed all occurrences of inversion to deconvolution.**

P9 L10: Are these data sources not redundant with the global maps of total and soil C storage described earlier?

**While the information of the global carbon content is also contained in the maps, we feel the inclusion of the additional, well established, global target is warranted by the importance of these targets. This is effectively increasing the weight of these targets.**

P9 L27: I don't understand the reasoning for the duplication of ensembles with gross transitions. Please motivate.

**We did not repeat the procedure to improve the prior distribution (See updated manuscript) for $M_{gross,gross}$ and as such the prior and posterior distributions do not converge. Consequently we do not feel comfortable to use $M_{gross,gross}$ as the basis for our estimates for $E_{LUC}$. As a compromise we introduced $M_{gross,net}$, retaining the confidence in the performance of $M_{net,net}$ and simply adding the important processes of shifting cultivation and wood harvest.**

P9 L31: As noted above, I have difficulties following this description.

**Section revised completely, see the answer to major points.**

P10 L5-8: Is material for the introduction, not the results section

**Removed Paragraph**

P10 L 8-11 can be safely removed.

**Removed Paragraph**

Section 3: When giving numerical estimates, please add either range or standard deviation, whenever the number is based on the ensemble. I also think that the more logical arrangement of the Results sections would be to first talk about model performance, and then about the attribution of the net land flux to LULCC and residual.

**Added the skill weighted 90% confidence interval for every reported number, except when reporting the median difference between two ensemble configurations. We agree that the suggested order of the result section is more logical, however we feel that the results on LULCC are of more interest to a broader range of readers, and thus prefer to lead with those results.**

P13 L4: Why would an underestimation of the ELUC not affect your conclusions about ELUC?

**The net land-atmosphere flux is underestimated because $\mathbf{E}_{gross,net}$ features additional processes that lead to an increase in $E_{\mathsf{LUC}}$, while the residual land sink remains constant. However if only considering $E_{\mathsf{LUC}}$ we expect the magnitude of the residual land-sink and net land-atmosphere flux to be less important than for instance model performance in the vegetation carbon benchmarks (Li et al., 2017). For clarity we revised the sentence:**
*"A caveat of this choice is that the net land-atmosphere flux is underestimated in $M_{gross,net}$ because the residual land sink only responds to the lower $E_{\mathsf{LUC}}$ of $M_{net,net}$. However if only considering $E_{\mathsf{LUC}}$ we expect the magnitude of the residual land-sink and net land-atmosphere flux to be less important than model performance with respect to vegetation carbon (Li et al., 2017) and other benchmarks."*

P17 L6: is the use of the word significant appropriate here?

**Changed 'significant' to 'relevant'**

P17 L8: why not?

**Using the vegetation carbon distribution directly, would have been a valid choice. Exchanging the total carbon distribution for the vegetation carbon distribution in the hierarchical weighting scheme reveals that the median parameter values used for the best guess version change less than 0.5%.**

P 18 L 3: Why is this different from the approach described in Section 2?

**Sentence shortened and clarified to read:**
*"We compare the total land-atmosphere exchange flux to the results of the atmospheric $CO_2$ deconvolution in Fig. 11"*

Conclusion Section: There is no need to repeat details of the methods or approach undertaken

**Shortened conclusion section by removing sentences which go into too much detail.**

Figure 1: Ensure all lines are visible

**Adjusted legend**

Table 1: Check units and definition for E0. This seems more like an activation energy to me (not a temperature sensitivity What are the units of the k_la:sa? Is this simply a scalar?

**$E_0$ is defined according to Lloyd and Taylor 1994, which considers the effect of an activation energy which is varying with temperature. Using their representation it has the unit [K] and is strictly speaking not an activation energy. As such we find the definition appropriate. $k_{la:sa}$ scales the PFT dependent leaf area to sapwood area and is as such unitless. The leaf area to sapwood area has the units [$m^2/m^2$]**

Please also note the supplement to this comment:
https://www.biogeosciences-discuss.net/bg-2018-62/bg-2018-62-AC1-supplement.pdf

―――――――――――――――――――――

[Figure]

[Figure]

**Fig. 1.** Median and 90% confidence intervals used for the prior distributions of the parameters of 9 ensembles. T1-T6 are ensembles with fewer members and E1 and E2 were precursors of the final ensemble E3.

**Supplement:**

[revised manuscript text omitted]

---

## Author Comment (AC2) · 6 Apr 2018

We want to thank the reviewer for the time and effort for the careful and very insightful review. In the following, we respond to the reviewer point by point, with our **responses in bold** and *quotations from the updated manuscript in cursive*. Please also consider the updated manuscript with track changes and a high-resolution figure for this reply in the supplementary. Also note that we expanded the discussion section to include a paragraph on a potential bias in the fossil fuel emissions used for the deconvolution by including non-fuel uses.

**Point-by-point response**

The study presents an approach to constrain a DGVM with multiple observational streams of carbon stocks, gross and net fluxes. The authors rely on a latin hypercube stratified sampling to perturb model parameters and create several 1,000-member ensemble simulations of the terrestrial carbon cycle for the historical period. Results focus on the estimation of land-use and land-cover change emissions. This study is quite innovative in the context of the global terrestrial carbon cycle as model parameters are constrained globally.

**Thank you**

I have found several similarities between the method described here and the Generalised Likelihood Uncertainty Estimation C1 method used in hydrological sciences (Beven and Binley 1992).

**We have added a reference to Beven and Binley in the introduction:**
*"Other approaches have also been investigated, such as using generalized likelihood function for model calibration and uncertainty estimation (Beven and Binley, 1992)"*

First, my main criticism targets the description of the sampling method. It is very unclear how the prior probability distribution in Figure 1 and the new best-guess values in Table 1 have been obtained, and how the posterior distribution of the parameters is calculated. Is it based on the selection criterion used to exclude the less skilled model parameters (p7 l5-8)? If Figure 1 and Table 1 present results from the current manuscript they should be described in the corresponding section.

**We restructured the method section and introduced a new subsection describing the explorative approach used to obtain the prior distribution. Additionally, we clarified the procedure to arrive at the posterior distribution in Section 2.3.2 (p.6 l.9-p.8 l.7 in the manuscript with track changes).**

**Figure 1 in this reply shows the evolution of the median parameter values and**

**ranges of the ensembles with 200 and 300 members (T1-T6) and large ensembles with 1000 members (E1-E3), discussed in the new section in the manuscript. Only the parameters used in the final ensemble E3 are shown. In the small ensembles, different model configuration and parameters were tested. For instance, in ensemble T2 nitrogen limitation was not considered and thus the nitrogen cycle related parameters were not sampled. The factorial simulation and small ensembles informed the choice for the prior of ensemble E1, which is then iteratively improved to arrive at the prior of E3.**

Second, I struggle to understand what experiments were actually undertaken. From section 2.6, it seems that three simulations are performed for each parameter set. These three simulations differ in the representation of LULCC: none (reference), gross or net transitions. Then, the results section reports the three different model configurations $M_{net,net}$, $M_{gross,net}$, $M_{gross,gross}$ while these are first described as three alternative skill weighted median.

**To clarify we added the following text to section 2.5 (Former 2.6):**
*"For each of the parameter sets 4 transient simulations over the industrial period are performed: (i) a simulation with prescribed net transitions ($M_{net,net}$ and $M_{gross,net}$), (ii) a simulation with prescribed gross transitions ($M_{gross,net}$ and $M_{gross,gross}$), (ii) a run with landuse area fixed at preindustrial levels and (iv) a run with landuse including shifting cultivation held at preindustrial levels. The last two simulations are used purely diagnostic to determine $E_{\mathsf{LUC}}$."*

Third, I am unclear about the skill-weighted mean method. Simulations with either net or gross land-use configuration are likely to yield different results so it is hard for me to justify $M_{gross,net}$. I understand that the $M_{gross,net}$ skill-weighted mean provides the best results compared to benchmarks (Table 3) but it could be an artefact, couldn't it?

**As now explained in the revised MS (see our answer above), we did not perform the procedure for optimizing the prior distribution for $M_{gross,gross}$. The prior and**

**posterior distributions of this configuration do not converge and as such we feel
not confident in using it as the basis for our estimates for $E_{\mathsf{LUC}}$. However, it is
clear from literature that processes such as shifting cultivation and wood har-
vest are an important component of landuse change. As a compromise we use
the optimized $\mathbf{M}_{net,net}$ ensemble with the additional gross transition processes
added, without the retuning of the model parameters.**

Also, some parameter sets are likely to perform better in some regions and worse in
other. Therefore, would a spatially-explicit weighting scheme (Schwalm et al., 2015;
Exbrayat et al. 2018) be more suited to constrain the ensemble?

**The use of spatially dependent parametrization offers numerous advantages,
which include the potential to yield better performance with regard to observa-
tional data. However we believe that assessing the performance of an individual
model using global parametrization, can still provide valuable insight in the ter-
restrial carbon cycle, as a potential caveat of regional parametrization are the
additional degrees of freedom which could potentially lead to an over-fitting of
the problem. We have added the following text to the discussion:**
*"An other avenue of increasing model performance is to introduce spatially explicit
parametrization, as recently used in multi-model averaging studies (Exbrayat et al.,
2018; Schwalm et al., 2015). A caveat of using this approach with a single model is a
potential overfitting of the parameters."*

Hereafter are some more specific comments p4 l6: CRU TS3.23 covers 1901-2014, so
how are simulations performed for 1800- 2014 (or is it 1800-2016 like in the abstract?)
please clarify throughout the manuscript

**Simulations are performed from 1800 to 2016 with recycled climate data from
1901-1930. Corrected wrong period 1901-2014 to 1901-2016 and changed the
wrong reference from CRU TS3.23 to CRU TS3.25 (1901-2016). The recycling of
the climate data is described at the end of section 2.2.**

p4 l21: please define what model metrics

**Added specification in sentence:**
*"..,the sampling is independent of the metrics used to assess model performance,.."*

p5 l6: how have these distributions been chosen?

**Please see update to sampling description.**

p7 l1: please write MSEi rel

**Done**

p9 l28: 'LULUC'? please correct here and in several other places

**Done, corrected LULUC to LULCC throughout the text**

p10 l13-23: please include some information about the uncertainty displayed in the Figures here and throughout the text

**We now report the skill weighted 90% confidence interval throughout the text, except for differences between different ensemble configurations.**

p11 l1: please quantify 'slight'

**The uptake from 1980 to 2016 amounts to 2.6 PgC. We now report the interval 1990-2016 and revised the sentence to read:**
*"The resulting total change in land carbon is negative, with a slight uptake of carbon at the end of the century, amounting to 9.3 (-0.9,22.2) PgC between 1990 and 2016"*

p14 l11: see previous comment on the study period

**The simulation spans 1800-2016, however spatial output was only saved after 1901 due to storage limitations.**

p16 l3: an informative figure would a covariance matrix of the parameter sets' scores for each criterion

**Figure 2 in this reply shows plots of the skill in individual observational targets for all parameter sets. A striking feature is the high correlation of the skill in total carbon map with skill in soil carbon map, which is not unexpected. There is no scatter plot with a lack of points in the upper right corner, i.e. no observational constraints are mutually exclusive. While we agree that this figure is informative, the sheer size and number of subplots make an inclusion in the manuscript or supplementary difficult. Please note that a version of the figure in vector graphic format is included in the supplementary of this reply.**

p16 l12: according to Figure 8b and d, the model captures the seasonality but not the interannual variability. This is worth reporting (and explaining).

**The interannual variability is not captured because the transport model used does not feature winds with interannual variability. Added sentence:**
*"As expected, the interannual variability in seasonal amplitude of $CO_2$ is not captured as the atmospheric transport model TM2 does not represent interannual variability in mass transport."*

Fig 1: Mnet,net is not defined

**Added:**
*"... ensemble with net land-use ($M_{net,net}$)"*

Fig 3: please explain the sign convention as it seems at odd with figure 4 (ELUC in particular)

**We updated Figure 4 to show a release of carbon to the atmosphere due to LULCC as positive, which is consistent with Figure 3 (And the rest of the text). Updated the figure caption and text to be consistent with this change.**

Fig 7: this figure is very complicated. Why is it important to look at the whole ensemble, and the constrained one? Constraining the ensemble uncertainty is not a major point in the rest of the manuscript and uncertainties are not reported in most of the text.

**We expanded the section explaining the constraining process (See answer above) and added additional confidence intervals for the numeric results. We revised the text in the first paragraph of 3.2 to better explain the figure:**

*In Fig. 7 a mapping of the $MSE_{rel}$ to an individual skill score is displayed for the observational data-sets with a spatial structure, to demonstrate how well the median of the ensemble and the new version LPX v1.4 are able to simulate individual observations. The figure also demonstrates the success of the assimilation process: the skill scores for many individual targets are improved in the ensemble median and LPX v1.4 compared to LPX v1.2, the starting point of our work. As a consequence of our iterative prior selection (section 2.3.2) the median skill for an individual constraint is similar in the constrained ensemble compared to the unconstrained ensemble. In all but the fAPAR benchmark the skill is consistently higher than the minimum skill criterion. With the exception of the biomass measurements by (Keith et al., 2009) and the fAPAR benchmark, the maximum skill in the constrained ensemble is identical to the full ensemble. The reduced maximum skill in those benchmarks is due to an exclusion of singular runs excelling at this benchmark but performing badly in others.*

Fig 8: please move the legend

**Done**

Please also note the supplement to this comment:
https://www.biogeosciences-discuss.net/bg-2018-62/bg-2018-62-AC2-supplement.zip
* * *
**Fig. 1.** Median and 90% confidence intervals used for the prior distributions of the parameters of 9 ensembles. T1-T6 are ensembles with fewer members and E1 and E2 were precursors of the final ensemble E3.

[Figure]

**Fig. 2.** Skill in observational targets for all parameter sets. The diagonal shows a histogram of the skills for the targets, the off-diagonal shows the skill of two observational datasets in a scatter plot.

---

## Author Comment (AC3) · 6 Apr 2018

We mistakenly quoted reviewer 2 (J.-F. Exbrayat) in the introductory statement, instead of reviewer 1 (S. Zaehle). We apologize for any confusion or inconvenience.

The paragraph should instead read:

"Lienert and Joos apply a bayesian data assimilation framework to the LPX-Bern model in order to constrain a selection of model parameters using a range of local to global carbon and water cycle observations. In the manuscript, they describe the framework and illustrate the key model performance criteria. This framework allows them to pro-

vide a data-constrained simulation of the regional and global terrestrial carbon balance between 1860 and 2016, and in particular to estimate the land-use related carbon emission, including an uncertainty range.

This is a very good study integrating multiple observations in a systematic and repro-ducable way to constrain a process-based global carbon cycle model. This system is not only used to produce a newly calibrated LPX-Bern version for future use, but also to provide useful insight into the magnitude (and particularly the uncertainty) of land-use emissions. Overall this is a valid contribution to Biogeosciences.

**Thank you.**"